# Exome-wide association study identifies *KDELR3* mutations in extreme myopia

Jian Yuan[1,2,7], You-Yuan Zhuang[1,7], Xiaoyu Liu[1,7], Yue Zhang[1,7], Kai Li[3], Zhen Ji Chen[1], Dandan Li[1], He Chen [4], Jiacheng Liang[1], Yinghao Yao[2], Xiangyi Yu[5], Ran Zhuo [1], Fei Zhao[1], Xiangtian Zhou [1,2], Myopia Associated Genetics and Intervention Consortium*, Xiaoguang Yu[5] ✉, Jia Qu[1,2,4] ✉ & Jianzhong Su [1,2,3] ✉

Extreme myopia (EM), defined as a spherical equivalent (SE) ≤ −10.00 diopters (D), is one of the leading causes of sight impairment. Known EM-associated variants only explain limited risk and are inadequate for clinical decision-making. To discover risk genes, we performed a whole-exome sequencing (WES) on 449 EM individuals and 9606 controls. We find a significant excess of rare protein-truncating variants (PTVs) in EM cases, enriched in the retrograde vesicle-mediated transport pathway. Employing single-cell RNA-sequencing (scRNA-seq) and a single-cell polygenic burden score (scPBS), we pinpointed *PI16 + /SFRP4+* fibroblasts as the most relevant cell type. We observed that *KDELR3* is highly expressed in scleral fibroblast and involved in scleral extracellular matrix (ECM) organization. The zebrafish model revealed that kdelr3 downregulation leads to elongated ocular axial length and increased lens diameter. Together, our study provides insight into the genetics of EM in humans and highlights *KDELR3*'s role in EM pathogenesis.

High myopia (HM), defined as a spherical equivalent (SE) of less than or equal to −6.00 diopters (D) or an axial length (AL) of more than or equal to 26.00 mm, has a high prevalence worldwide, particularly in Asian, and is one of the major causes of blindness[1]. An SE of less than or equal to −10.00 D, or AL of more than or equal to 28.00 mm are diagnosed with extreme myopia (EM)[2,3]. High and extreme myopia will increase the risk of various pathologic ocular complications[4]. The scleral and choroidal tissues in patients with EM often undergo degeneration and become significantly thinner, which are associated with excessive elongation of AL, vitreous degeneration, retinal detachment, and macular degeneration, thus greatly increasing the risk of blindness[5–8]. Moreover, the density of parapapillary blood vessels and the average sensitivity of the visual field are significantly decreased in EM, which can lead to retinopathy and optic neuropathy[9–11]. Current clinical technology cannot provide an effective approach to radically slow or stop the inappropriate growth of AL. Additionally, EM is always accompanied by irreversible pathologic or degenerative alterations in ocular tissues, there are likewise no effective interventions. After excimer laser surgery in patients with EM, the prognosis is shown to be less pronounced than in myopia in terms of effectiveness and safety[12], and scleral buckling alone in retinal detachment patients with EM is also less effective than in retinal detachment patients with HM[13]. Therefore, exploring the pathogenesis of EM is important for early diagnosis and efficient interventions in patients with EM.

As an extreme phenotype of HM, there is substantial evidence to suggest that EM is primarily associated with genetic factors rather than being solely caused by behavioral or environmental factors, such as

[1]National Engineering Research Center of Ophthalmology and Optometry, Eye Hospital, Wenzhou Medical University, Wenzhou, China. [2]Oujiang Laboratory, Zhejiang Lab for Regenerative Medicine, Vision and Brain Health, Wenzhou, Zhejiang, China. [3]Wenzhou Institute, University of Chinese Academy of Sciences, Wenzhou, China. [4]School of Biomedical Engineering, Hainan University, Haikou, China. [5]Institute of PSI Genomics, Wenzhou, China. [7]These authors contributed equally: Jian Yuan, You-Yuan Zhuang, Xiaoyu Liu, Yue Zhang. *A list of authors and their affiliations appears at the end of the paper. ✉e-mail: yuxiaoguang@psi-gene.com; qujia@eye.ac.cn; sujz@wmu.edu.cn

long-time near work alone. For instance, an epidemiologic study that observed a consistent prevalence of EM across every age range supports the notion that genetic factors exert a greater influence on the development of EM[14]. In addition, a series of genetic association studies have found that numerous single-nucleotide variants (SNVs) in genes related to refractive errors are associated with EM, rather than common myopia. Xu et al.[15] found that the rs4575941 in the *SOX2* gene was associated with susceptibility to EM in a Chinese Han population. Variants rs13382811 in *ZFHX1B*, rs7839488 in *SNTB1*, and rs644242 in *PAX6* were reported only significantly associated with EM[16,17]. Scleral hypoxia is believed to play a crucial role in the interactions between genetic and environmental during the development of myopia[18]. A genome-wide association study (GWAS) has demonstrated that the HIF-1α signaling pathway is enriched with more genetic variants in patients with EM but not in HM[19]. This suggests that genetic variations in the HIF-1α signaling pathway are likely to contribute to the development of EM. All of the aforementioned studies collectively support a molecular genetic-dominated pathogenesis of EM. However, previous trio and family-based exome sequencing studies in individuals with monogenetic forms of EM have focused on several rare coding variants in known myopia genes, which can only explain a small fraction of EM cases in the general population. Hence, large-scale population genetic data are of utmost importance in our quest to unveil the genetic etiology and pathogenesis of EM.

In this study, we performed the largest EM exome case-control study to date, involving 449 cases and 9606 controls, and followed by a series of bioinformatic analyses in combination with experimental exploration. We observed an excess of rare PTVs in EM cases, and these variants primarily resided in genes involved in protein secretory pathways. We identified variants in the *KDELR3* locus, which result in elongated eye axis and increased lens diameter, as the major contributors to EM. Furthermore, we introduced the single-cell polygenic burden score (scPBS), a method to assess cell-type-disease association through integrated scRNA-seq and rare-variant polygenic burden test, and found fibroblasts from choroid and sclera were associated with EM.

## Results

### Exome sequencing identifies diagnostic mutations in EM cases

After stringent quality control, we used WES data from 449 EM-affected individuals and 9606 healthy controls in the discovery stage (Fig. 1a, Supplementary Data 1). A total of 2,543,192 variants were used for further analysis, including 50,470 common variants (minor allele frequency, MAF > 5%), 40,541 low-frequency variants (0.5% < MAF < 5%), and 2,452,077 rare variants (MAF < 0.5%) (Supplementary Data 2). We first quantified the contribution yield (defined as the percentage of cases) to EM attributable to potential pathogenic variants in 75 well-characterized myopia-associated genes (Supplementary Data 3). Variant pathogenicity in these known genes was evaluated by manual review following guidelines of the American College of Medical Genetics and Genomics (ACMG) (Supplementary Data 4). Among 2684 variants detected in known genes, 27P/LP variants were identified across 21 known genes (Supplementary Fig. 1), including 20 (74.1%) PTVs and 7 (25.9%) missense variants. Among the 27P/LP variants, 13 (48.1%) of which were absent in the gnomAD database, only a variant had MAF > 0.05%; all of the variants had PHRED-scaled CADD scores of greater than 20. Among those genes, *CEP290* accounted for the highest proportion of pathogenic alleles in cases (AC = 12, Fig. 1b), which was reported to cause Bardet–Biedl syndrome [MIM: 615991] and Leber congenital amaurosis [MIM: 611755] with early-onset retinal degeneration[20].

The P/LP variants in known myopia genes were detected in 41 patients, yielding a 9.13% contribution to our EM cohort (Fig. 1c), among which most (39/41, 95.12%) carried monoallelic is single

heterozygous-P/LP variants (Fig. 1d). *CEP290*, *RP1*, and *LTBP2* are the top three potential EM disease-causing genes (>5%) in our cohort (Fig. 1e). Systemic syndromic, eye syndromic and non-syndromic myopia-associated genes accounted for 31.70%, 60.98% and 7.32% in genetic detection, respectively (Fig. 1f). Using functional enrichment analysis, we classified the associations between known myopia genes and the biological processes defined by gene ontology. Visual perception (GO:0007601; $P = 3.69 \times 10^{-49}$), eye development (GO:0001654; $P = 4.54 \times 10^{-17}$), extracellular matrix organization (R-HSA-1474244; $P = 2.15 \times 10^{-14}$), and retina homeostasis (GO:0001895; $P = 1.30 \times 10^{-9}$) were enriched, which may contribute to different mechanisms of EM (Supplementary Data 5). Of note, 9 genes implicated in eye development accounted for the largest proportion (23/41, 56.10%) of detected cases, including *CEP290*, *RP1*, *EPHA2*, *BBS7*, *WDPCP*, *PAX2*, *COL2A1*, *COL18A1*, and *SCO2* (Fig. 1g). Besides, 6 genes were responsible for extracellular matrix organization (*EPHA2*, *COL9A2*, *PAX2*, *COL2A1*, *LTBP2*, and *COL18A1*) and also comprised a sizable proportion (10/41, 24.39%) of known genes (Fig. 1g). Consistent with previous research findings, genes expressed in the sclera and 9 myopia-associated genes expressed in photoreceptors (cone and rod cells) with a proportion of 21.95% and 56.10% in the solved EM cases, respectively (Fig. 1h).

### Excess of exome-wide and gene-set-based rare PTVs

We tested all 91,011 common and low-frequency variants (MAF > 0.5%) for EM association using a linear mixed model-based association analysis method (EMMAX and MLMA-LOCO) that can account for population structure and relatedness. There was no significant evidence of association with EM for sets of common or low-frequency variants at the exome-wide significance level of $P < 4.3 \times 10^{-7}$ for coding variants[21] (Supplementary Fig. 2). Therefore, we adopted a complementary strategy, focusing on variants in protein-coding sequence, and sought to improve power to detect rare-variant association by exploiting the more robust functional annotation of coding variation and the potential to aggregate multiple alleles of presumed similar impact in the gene set or same gene.

We first evaluated the association of the burden of all rare PTVs, D-mis, B-mis, or synonymous variants with HM by three logistic models, and then dissected the burden test into those focusing on specific gene sets (Supplementary Fig. 3, Supplementary Data 6). Specifically, each model used Firth-based logistic regression and incorporated some or all of the following covariates: (1) sample sex, (2) principal component (PC) 1–10 and either (3) the total exome count (summation of synonymous variants, B-Mis, D-Mis and PTVs). Using these models, we evaluated the excess of exome-wide rare variants in primary cohorts (449 cases and 9606 controls) and down-sampling cohorts (449 cases and 449 controls) due to logistic regression that relies on asymptotic properties. According to the relatively most conservative model which included all the covariates (sex, PC1-PC10, and total exome count), we observed a significant enrichment of rare PTVs in primary cohorts (OR = 1.08, $P = 4.61 \times 10^{-6}$, Fig. 2a), but not of D-mis. There was no significant difference in excess rare synonymous variants and B-mis between cases and controls, which can be considered as a negative control, suggesting that enrichment of potential pathogenic variants is unlikely to be due to un-modeled population stratification or technical artifact.

Biologically informed gene sets can refine our understanding of how rare PTVs confer risk for EM and generate potential biological hypotheses for follow-up analyses. To attempt to elucidate pathways enriched for damaging variation in individuals with EM in an agnostic manner, we performed a collapsing analysis using Fisher's exact test on 10,587 gene sets derived from Human MSigDB Collections[22] (Supplementary Fig. 4). Among the 10,587 gene sets, we observed significant enrichment of rare PTVs in two gene sets in EM cases, including Retrograde vesicle-mediated transport, Golgi to endoplasmic reticulum

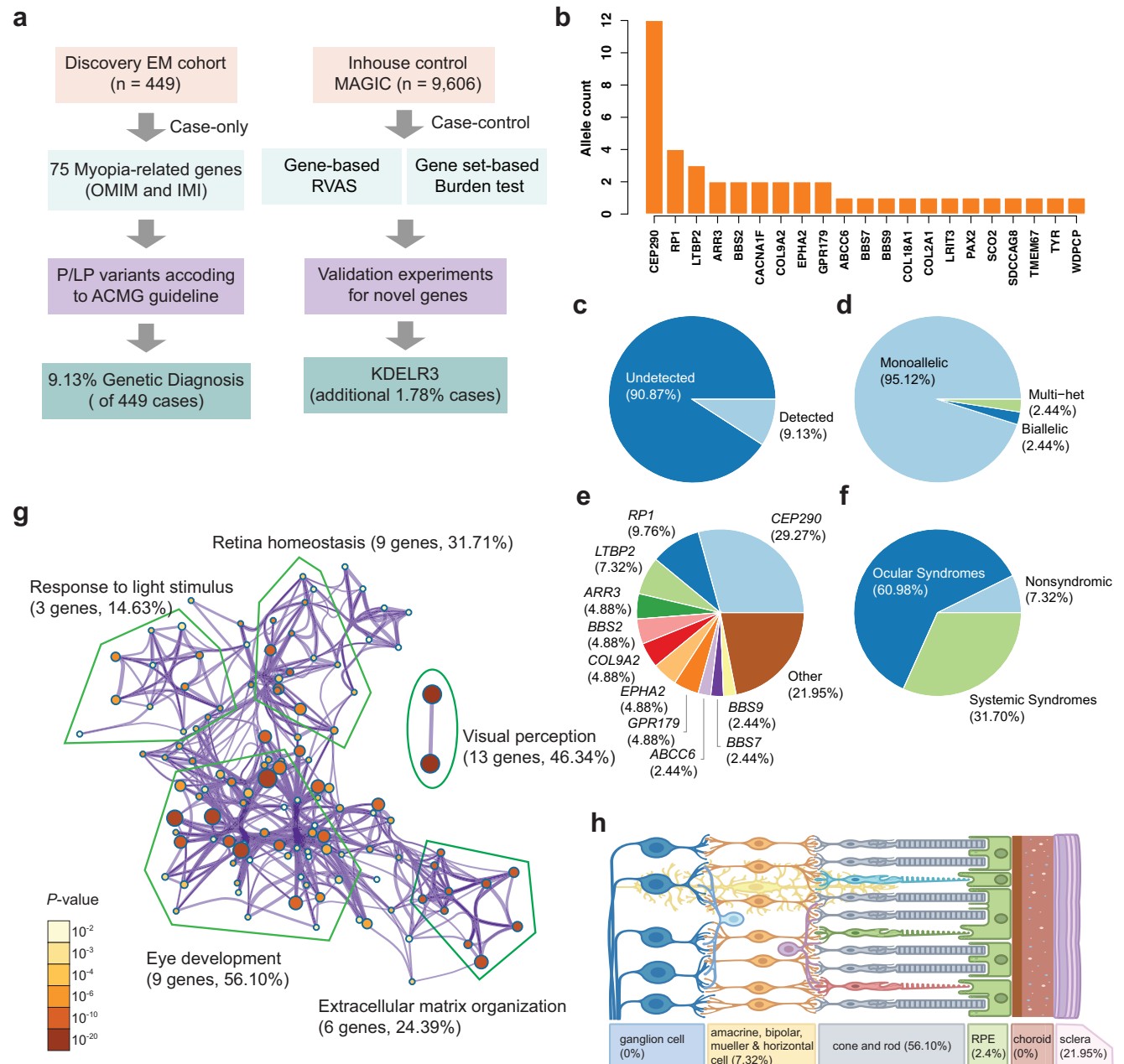

**Fig. 1 | Study flowchart and overview of P/LP variants in known EM genes.**
**a** Flowchart for the study design to characterize and identify deleterious variants in known and unreported EM genes. A total of 449 EM patients were recruited for visual acuity, autorefraction testing, and whole-exome sequencing. **b** Allele counts of P/LP variants detected in 33 of 75 known EM genes. **c** Diagnostic yield of known EM-causing genes in 449 patients. **d** The proportion of each mode of inheritance in the 102 patients carrying P/LP variants in known EM genes. **e** The proportional contribution of each gene among 102 cases. **f** The proportion of patients classified

according to the clinical syndromic information of the affected genes. **g** GO pathway enrichment for 33 known EM genes. This network shows the terms with a P-value < 0.01 identified by Metascape, a minimum gene count of 3, and an enrichment factor >1.5. The nodes sizes are scaled with a P-value. The known EM genes are most significantly enriched in visual perception, eye development, and extracellular matrix organization. The number in parentheses shows the gene number and proportional contribution of each gene. **h** Schematic overview of proportional contribution of retinal cell-specific genes.

(GO:0006890, OR = 5.94, $P = 2.17 \times 10^{-6}$, FDR = 0.023) and Unfolded protein response (R-HSA-381119, OR = 3.70, $P = 1.78 \times 10^{-5}$, FDR = 0.047), after correcting for multiple tests (Supplementary Data 7). The pathway collapsing analyses remained significant when tested against an empirical distribution generated by random sampling of the same number of length-matched genes at random 1000 times (Supplementary Data 7). No significant pathway enrichment was observed for D-mis, B-mis, and synonymous variants, which can be considered as a negative control (Supplementary Data 7–10). Consistently, we identified significant enrichment of rare PTVs in

Retrograde vesicle-mediated transport, Golgi to endoplasmic reticulum (OR = 4.41, $P = 1.06 \times 10^{-5}$) among EM cases compared to controls under the most conservative model used Firth-based logistic regression (Fig. 2b).

## Inferring relevant cell types for EM by using single-cell gene expression
Our next goal was to use the polygenic burden of rare variants to identify EM-relevant cell types. To robustly identify cell type implied by EM, we used two approaches that link scRNA-seq with the polygenic

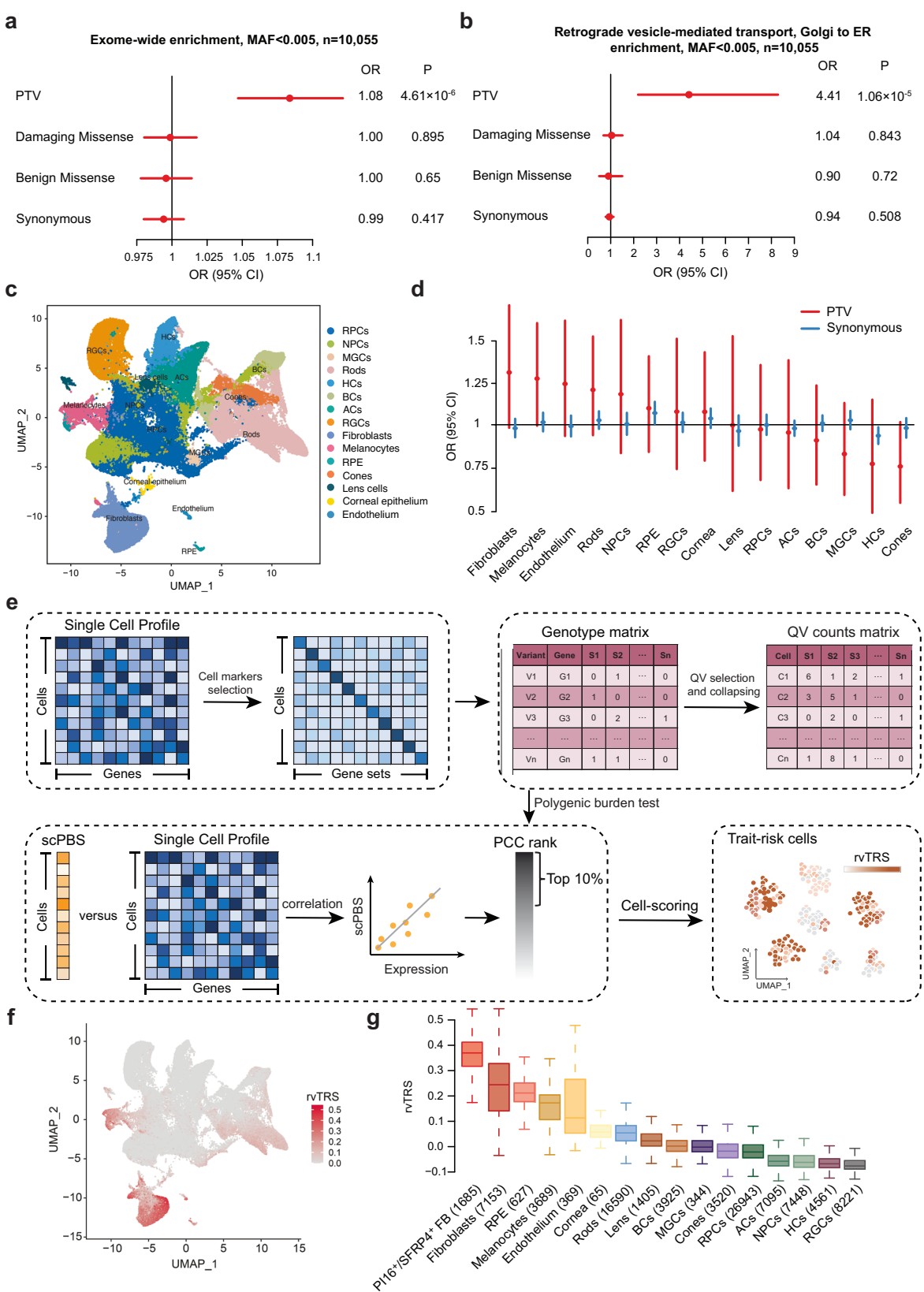

**a** Exome-wide enrichment, MAF<0.005, n=10,055

**b** Retrograde vesicle-mediated transport, Golgi to ER enrichment, MAF<0.005, n=10,055

burden of rare variants in EM at predefined cell-type resolution and single-cell resolution. We leveraged gene expression data from a wide spectrum of cell types from the human embryonic eyes[23] to systematically map EM to cell types (Fig. 2c). For the cell-type level, we tested whether the top 10% marker genes in each cell type were enriched burden of rare PTVs in EM cases (Supplementary Data 11). We found

the strongest signals for fibroblasts ($P = 0.050$) and melanocytes ($P = 0.042$) were nominal significantly associated with EM (Fig. 2d, Supplementary Data 12). Significant associations with neurons and glial cells were not observed. The burden test of synonymous variants showed that there was no significant inflation of background rate between the case and control cohorts. In conclusion, we have shown

**Fig. 2 | Exome-wide, gene set, and single-cell polygenic burden test of PTVs in EM cases. a** Exome-wide burden analysis of PTVs, D-mis, B-mis, and synonymous variants within rare variants (MAF < 0.5%). Significance of association is displayed with the *P*-values from the two-sided Firth logistic regression test; errors bars indicated 95% confidence intervals (CIs) of the corresponding odds ratios. Multiple test correction *P* = 0.01. The graphs display the means and standard deviations. **b** Burden analysis of genes involved in retrograde vesicle-mediacted transport, Golgi to ER. PTVs, D-mis, B-mis, and synonymous variants within rare variants are displayed. Significance of association is displayed with the *P*-values from the two-sided Firth logistic regression test; errors bars indicated 95% confidence intervals (CIs) of the corresponding odds ratios. Multiple test correction *P* = 0.01. **c** Uniform manifold approximation and projection (UMAP) embedding plot shows the cellular component of a scRNA-seq dataset for human embryonic eyes. RPCs retinal progenitor cells, NPCs neurogenic RPCs, MGCs Müller glial cells, HCs horizontal cells,

BCs bipolar cells, ACs amacrine cells, RGCs retinal ganglion cells, RPE retinal pigment epithelia, Endothelium endothelial cells. **d** Burden analysis of genes highly expressed in each cell types (testing for enrichment in rare variants of the 10% most specific genes in cell type). PTVs and synonymous variants within rare variants are displayed. The sample size for EM is as follows: Ncases = 449, Ncontrols = 9,606. The black solid line represents the null (OR = 1). Each point shows the point estimate of OR from logistic regression. Bars show 95% confidence intervals (CI). **e** Overview of scPBS approach. scPBS takes a case-control genotype matrix and a scRNA-seq data set as input and outputs individual-cell-level polygenic burden score for association with the disease. **f** Per-cell rvTRS calculated by scPBS and Seurat for EM are shown in tSNE coordinates. **g** Per-cell types rvTRS. Boxplots show the median with interquartile range (IQR) (25%–75%); whiskers extend 1.53 the IQR. The number of cells in each cell types is shown in parentheses. Source data are provided as a Source Data file.

that cell-type level gene expression allows the identification of relevant cell types for EM.

To characterize considerable heterogeneity within each cell type, we applied the single-cell enrichment approach MAGMA-based scDRS[24], to discern EM-associated cell types and subpopulations with above ExWAS. However, no genes were significantly associated with the HM in MAGMA analysis (Supplementary Fig. 5). scDRS did not identify the EM-relevant cells with significant individual-cell-level disease association (Supplementary Fig. 6). Therefore, we introduce the single-cell polygenic burden score (scPBS) in our MAGIC toolkits (https://github.com/sulab-wmu/MAGIC), a method to evaluate polygenic burden enrichment of rare variants in individual cells of scRNA-seq data. scPBS assesses whether a given cell has an excess of rare PTVs among EM cases in cell-specific highly expressed genes derived from scRNA-seq. scPBS consists of three steps, which output a rare-variant-based trait-relevant score (rvTRS) of each cell computed by averaging the expression level of the trait-relevant genes (Fig. 2e, "Methods" section). The rvTRSs of cells from the human embryonic eyes for EM are shown in low-dimensional Uniform manifold approximation and projection (UMAP) space (Fig. 2f). Remarkably, cell lineages relevant to EM yielded considerably high rvTRSs, indicating that the cell specificity of these rare variants genetic effects was well captured by scPBS (Fig. 2f). Consistent with above predefined cell-type resolution, we found scPBS identified fibroblasts compartment with increased rvTRSs. The subpopulation of EM-associated fibroblasts had high expression of matrix fibroblasts cell marker genes (*PI16*+ and *SFRP4*+)[25,26] (Supplementary Fig. 7, Supplementary Data 13). scPBS identified *PI16*+/*SFRP4*+ fibroblasts exhibited increased rvTRSs for EM risk (Fig. 2g). Together, these findings provide genetic evidence supporting the involvement of fibroblasts subpopulation, namely *PI16*+/*SFRP4*+ fibroblasts, in the etiology of EM disease.

## Gene-based rare-variant association analysis

To identify genes associated with EM, we performed an association analysis where individuals were categorized based on the presence or absence of rare deleterious variants with MAF < 0.5%, considering both our cohorts and data from gnomAD. The rare PTVs model included 12,384 variants across 6,335 genes identified in the total cohort (cases and controls). We thus set the exome-wide significance threshold at Bonferroni corrected *P*-value of $0.05/6335 = 7.89 \times 10^{-6}$ by using two-sided FET *P*-values. The genes significantly associated with EM included *KDELR3* (OR = 15.82, $P = 7.27 \times 10^{-7}$) and *VN1R4* (OR = 10.89, $P = 1.49 \times 10^{-6}$; Fig. 3a, Supplementary Data 14). The quantile–quantile plot demonstrated good control of systematic bias and deviation from the null hypothesis only at the lowest *P*-values (Supplementary Fig. 8). To validate the top-ranked genes with EM relationships, we conducted enrichment analysis between the top 100 candidate genes and disease-related genes from DisGeNET[27], and found 393 Blindness-related genes were significantly enriched in the top 100 candidate genes (5.4-fold

enrichment, *P* = 0.00032; Supplementary Data 15). The top 10 genes in the collapsing analysis are listed in Fig. 3b. The strongest signal in the collapsing analysis was generated by variants in the *KDELR3* locus, which 8 of 449 cases had three rare PTVs (1.8%), whereas only 11 of 9606 controls had a rare PTV (0.1%; Table 1). There were no cases had rare PTV in *KDELR3* or P/LP variants in known myopia genes (Supplementary Fig. 9). In order to compare case-control burden while considering the potential mixture of effect sizes and directions resulting from rare coding variation, as well as the influence of 10 PCs of population stratification, we performed the SKAT-O for rare-variant association study and found *KDELR3* had exome-wide significant signals ($P = 7.42 \times 10^{-12}$; Supplementary Data 14).

The *KDELR3* gene, located on chromosome 22, comprises 5 exons, each spanning more than 1694 base pairs. It encodes KDEL endoplasmic reticulum protein retention receptor 3. The KDEL receptors (KDELRs) are a family of seven-transmembrane-domain ER protein retention receptors that function in Golgi transport control and the ER quality control[28,29]. These signaling pathways coordinate membrane trafficking flows of ECM proteins[18] and control ER stress response[30], an important step in myopia progression. Similarly, gene functional enrichment analysis with the top 100 EM genes identified 6 genes in the "Extracellular matrix organization" category (R-HSA-1474244, 6.06-fold enrichment, *P* = 0.00049) and 3 genes in the "cell-substrate junction assembly" category (GO:0009987, 22.73-fold enrichment, *P* = 0.00032; Supplementary Data 16). We found *KDELR1*, *KDELR2*, and *KDELR3* are members of genes involved in Retrograde vesicle-mediated transport, Golgi to endoplasmic reticulum, which is enriched rare PTV among EM cases. When removed *KDELR3* from gene sets, no significant enrichment was detected (OR = 2.23, *P* = 0.112; Supplementary Fig. 10). Considering the basic cellular functions performed by KDELRs, their association with EM is not surprising. We observed that all three deleterious mutations occurred within a highly conserved domain (Fig. 3c). Comparative sequence analysis shows that all three of the rare PTVs are located in sites that are highly conserved between species, including both vertebrates and invertebrates (Fig. 3d). We further investigated the association of *KDELR3* PTVs with lower SE in our MAGIC cohorts[31]. Heterozygous carriers of *KDELR3* PTVs was significantly associated with lower SE of 0.75 D, and their distribution across SE diopters was drastically shifted toward lower SE diopters in the both eyes (*P* = 0.01799 for right and *P* = 0.00040 for left, respectively, Wilcoxon rank sum test; Fig. 3e). We also performed Fisher's exact test to analyze the association of rare PTVs in *KDELR3* with high myopia (−10.00 D ≤ SE ≤ −6.00 D) and found a nominal significant association between them (OR = 2.19, *P* = 0.038). Based on bulk RNA-sequencing (RNA-seq) data of 55 different tissues from the Genotype-Tissue Expression (GTEx) consortium[32], we found that *KDELR3* was significantly high expressed in fibroblasts than the other tissues (Fig. 3f). Cell-type specificity analysis of data from human choroid and sclera cells[33,34] consistently showed that *KDELR3* was mainly expressed in ocular fibroblasts (Fig. 3g), suggesting that

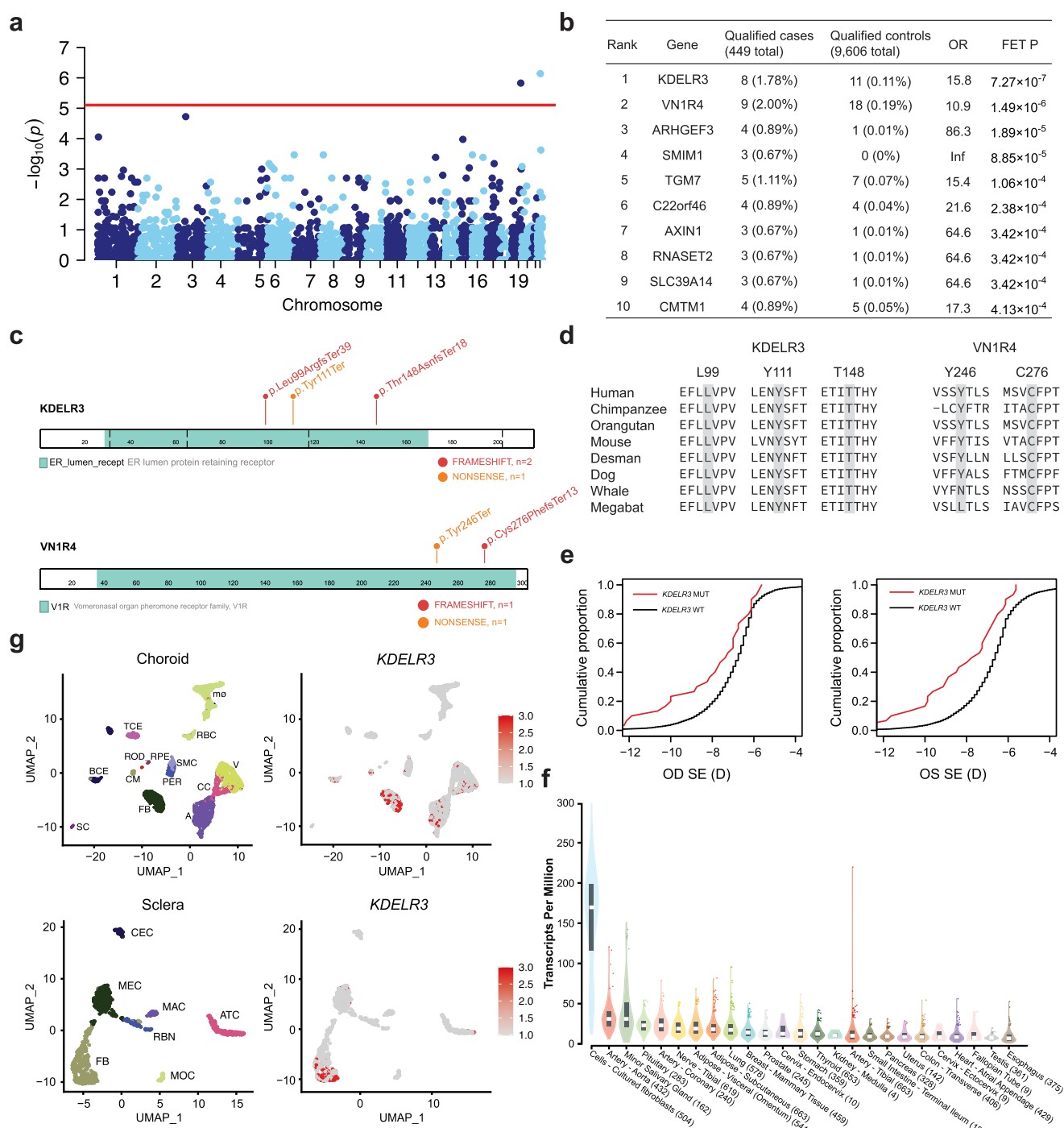

**a** Manhattan plot of gene-based collapsing analysis

**b**

| Rank | Gene | Qualified cases (449 total) | Qualified controls (9,606 total) | OR | FET P |
|------|------|------|------|------|------|
| 1 | KDELR3 | 8 (1.78%) | 11 (0.11%) | 15.8 | $7.27 \times 10^{-7}$ |
| 2 | VN1R4 | 9 (2.00%) | 18 (0.19%) | 10.9 | $1.49 \times 10^{-6}$ |
| 3 | ARHGEF3 | 4 (0.89%) | 1 (0.01%) | 86.3 | $1.89 \times 10^{-5}$ |
| 4 | SMIM1 | 3 (0.67%) | 0 (0%) | Inf | $8.85 \times 10^{-5}$ |
| 5 | TGM7 | 5 (1.11%) | 7 (0.07%) | 15.4 | $1.06 \times 10^{-4}$ |
| 6 | C22orf46 | 4 (0.89%) | 4 (0.04%) | 21.6 | $2.38 \times 10^{-4}$ |
| 7 | AXIN1 | 3 (0.67%) | 1 (0.01%) | 64.6 | $3.42 \times 10^{-4}$ |
| 8 | RNASET2 | 3 (0.67%) | 1 (0.01%) | 64.6 | $3.42 \times 10^{-4}$ |
| 9 | SLC39A14 | 3 (0.67%) | 1 (0.01%) | 64.6 | $3.42 \times 10^{-4}$ |
| 10 | CMTM1 | 4 (0.89%) | 5 (0.05%) | 17.3 | $4.13 \times 10^{-4}$ |

**Fig. 3 | Collapsing analysis identifies *KDELR3* as an EM gene. a** Manhattan plots of the gene-based collapsing analysis. An excess of rare PTVs in EM cases within genes were tested using Fisher's exact test [FET], Red line, $P = 0.05/6335 = 7.89 \times 10^{-06}$. **b** The top ten genes from collapsing analyses under the same model are shown, including the exact numbers of all qualifying cases and controls and statistical calculations of association (OR and FET *P*). **c** Linear schematic of the *KDELR3* and *VN1R4* encoded protein and location of genetic variants identified by WES. Frameshift variants are in red, Nonsense variants in orange. **d** Multiple sequence alignment indicating a degree of sequence conservation across species at the locations of *KDELR3* and *VN1R4* rare PTVs. **e** Cumulative distribution of SER in MAGIC cohort. The x-axis is the SER of right (OD) and left (OS) eyes. The y-axis is the cumulative fraction of individuals. red: individuals carrying *KDELR3* PTVs, black:

individuals without carrying *KDELR3* PTVs. **f** Violin plots indicate the expression of *KDELR3* genes retina and other tissues in GTEx. Sample size for each tissue is shown in parentheses. The center line in the box plots contained within each violin plot shows the median, the box edges depict the interquartile range (IQR), and whiskers mark 1.5× the IQR. The violin plot edges represent the minima and maxima values. **g** UMAP of all tissues single-cell data with cell colored based on the expression of *KDELR3* genes for particular cell types. Gene expression levels are indicated by shades of red. SC Schwann cells, BCE B-cell, TCE T-cell, CM melanocytes, FB fibroblast, RPE retinal pigment epithelium, SMC smooth muscle cells, PER pericytes, RBC rod bipolar cells, ROD rods, mo macrophage, A artery, CC choriocapillaris, V vein, MOC monocytes, RBN retinal bipolar neuron, MEC melanocytes, MAC macrophage, ATC active T cells, CEC corneal epithelial cells.

**Table 1 | All qualified variants in KDELR3 in 449 EM cases and 9,606 controls with gnomAD MAF < 0.5%**

| Variant ID | HGVS.c | HGVS.p | Mutation type | Alleles in 449 HM cases (n) | Alleles in 9606 controls (n) | MAF in gnomAD (if available) | LOFTEE |
|---|---|---|---|---|---|---|---|
| **KDELR3** | | | | | | | |
| chr22-38875700-CT-C | c.296del | p.Leu99ArgfsTer39 | frameshift | 6 | 6 | 0.0001631 | HC |
| chr22-38875736-TAC-T | c.333_334del | p.Tyr111Ter | frameshift | 1 | 3 | 0.0003508 | HC |
| chr22-38877305-T-TA | c.442dup | p.Thr148AsnfsTer18 | frameshift | 1 | 2 | 0.0001631 | HC |
| **VN1R4** | | | | | | | |
| chr19-53770092-CAT-C | c.825_826del | p.Cys276PhefsTer13 | frameshift | 9 | 18 | 0.001604 | LC |
| chr19-53770181-G-T | c.738C>A | p.Tyr246Ter | stop gained | 1 | 0 | - | LC |
| chr19-53770281-A-AC | c.637dup | p.Val213GlyfsTer5 | frameshift | 0 | 1 | - | LC |
| chr19-53770460-C-T | c.459G>A | p.Trp153Ter | stop gained | 0 | 1 | - | LC |

deleterious mutation in *KDELR3* might affect scleral fibroblasts secrete collagen and other ECM components.

**Functional analysis of *KDELR3* variants in human and zebrafish**

We conducted a comprehensive ophthalmologic examination on six randomly selected EM eyes, six EM eyes from individuals who carried *KDELR3* mutations, and six eyes from individuals with normal visual acuity as a control group. Subfoveal retinal thickness and subfoveal choroidal thickness were measured using the SS-OCT (Fig. 4a). Overall, EM cases and healthy controls presented significant differences ($P < 0.05$) in terms of spherical equivalent refraction, astigmatism, axial length and intraocular pressure (Fig. 4b). For subfoveal choroidal thickness, a significant difference was found between the EM cases with or without *KDELR3* mutations and controls ($P = 0.009$ and $P = 0.002$; Fig. 4b); however, there is no significant difference of the subfoveal retinal thickness between the EM cases and control groups. In the subgroup analysis between EM cases with *KDELR3* mutation and randomly selected EM cases, limited by the sample size, there was no significant difference. A large numerical difference in subfoveal choroidal thickness between EM cases with *KDELR3* mutation (283.2 um) and randomly selected EM cases (338.8 μm) can be observed (Fig. 4b).

To further investigate the biological features and the effect on ocular refractive development on disruption of the *kdelr3* gene, we created a *kdelr3*-deficient zebrafish model through the use of morpholino oligonucleotide (MO)-induced knockdown. We firstly tested *kdelr3* spatiotemporal expression patterns of *kdelr3* during early embryogenesis in zebrafish. qRT-PCR showed that *kdelr3* expression decreased markedly from 1 to 7 days post fertilization (dpf) and exclusively expressed in the kidney, liver, brain, and eye (Supplementary Fig. 11). To test the function of the *kdelr3* gene in zebrafish eyes, we used MO that targets the splice sites of the *kdelr3* gene to construct a knockdown model. Four different concentrations of *kdelr3* MOs, together with a standard control MO, were microinjected into the yolks of embryos, respectively (Supplementary Fig. 12). Next, we used the electrophoresis analysis of RT-PCR from *kdelr3*-MO zebrafish of 3dpf to 5dpf and found that the knockdown of kdelr3-MO was effective and enabled to maintain up to 5 dpf (Supplementary Fig. 13). In addition, we used both SUPPA2[35] and custom analysis scripts to identify transcripts associations of MO-induced skipped exon in our RNA-seq datasets with *kdelr3*-MO zebrafish eyeballs, and found that *kdelr3*-MO zebrafish exhibited aberrant use of exons compared with the control zebrafish eyeballs (Supplementary Fig. 14, Supplementary Data 17). Strikingly, both the low-concentration knockdown groups (injected with 0.15 and 0.25 ng) exhibited evident myopia-related characteristics, such as an elongated eye axis and lens diameter (Fig. 4c). However, as the MO dosage was further increased, the zebrafish showed deformities with shortened eye axis and lens diameters. Nevertheless, the ratios of eye axis length to body length and lens size to body length remained elevated. To further verify that the ocular phenotype found in kdelr3-deficient morphants is not due to off-target effects, we performed mRNA compensation rescue experiments in rescue experiments. We found that the phenotype of the *kdelr3* knockdown is reversible after using full-length mRNA compensation in *kdelr3*-deficient morphants (Supplementary Fig. 15). To determine the retinal structural changes of *kdelr3*-knockdown morphants, we used the specific Tg(kdrl:mCherry), Tg(gad1b:mCherry) and Tg(gfap-eGFP) zebrafish lines and markers Anti-Recoverin RCVRN to observe blood vessel endothelial cell, retinal pigment epithelium cell, müller cell and photoreceptors, respectively (Fig. 4e). Compared to zebrafish received control MO, zebrafish that received 0.25 and 0.5 ng *kdelr3*-MO displayed significant reduction of blood vessel endothelial cell and photoreceptors in the retina, while retinal pigment epithelium cell and müller cell was not altered (Fig. 4d, e). Notably, the blood vessel endothelial cell signals of the 0.5 ng *kdelr3*-MO group decreased to

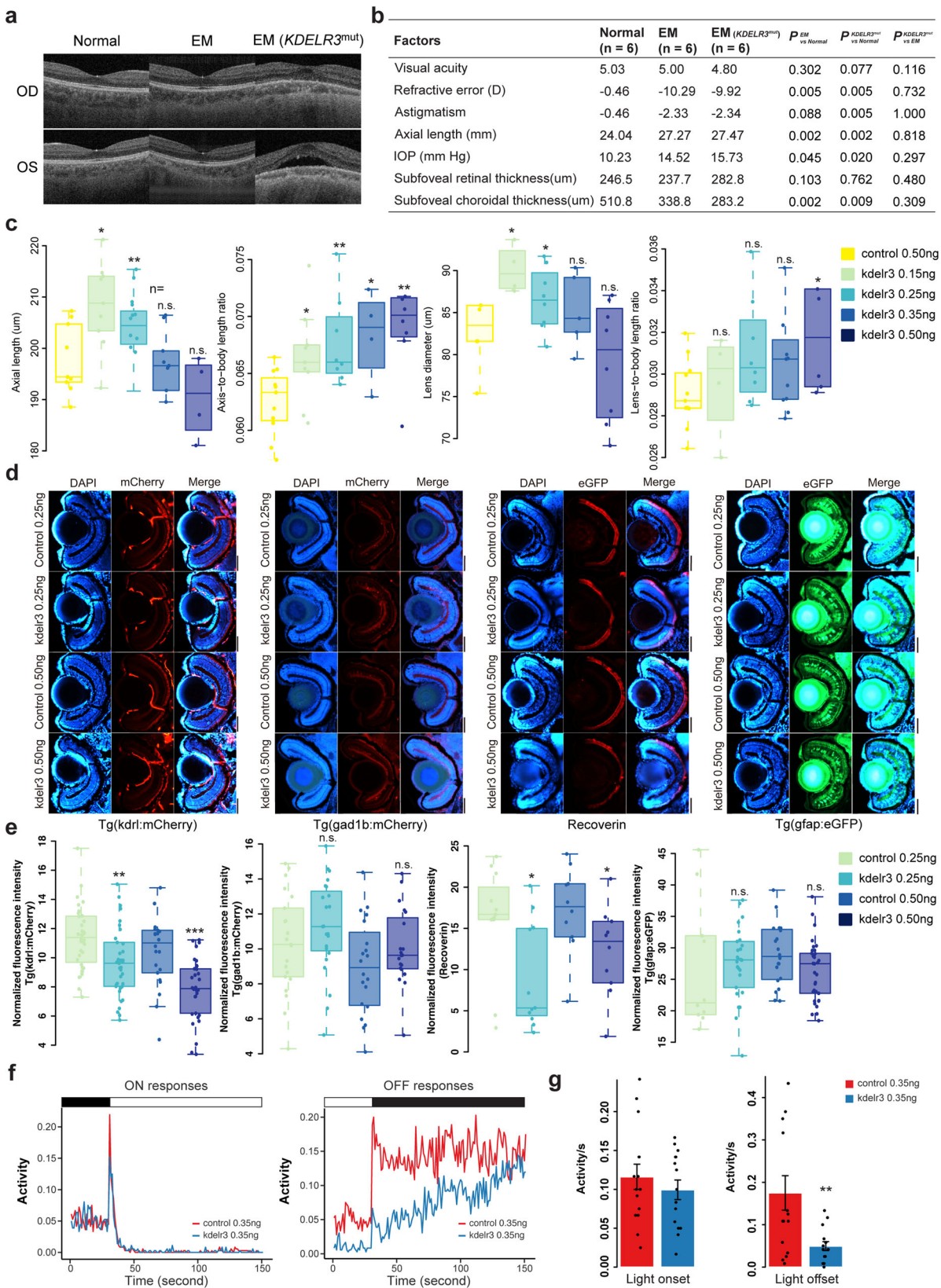

| Factors | Normal (n = 6) | EM (n = 6) | EM (KDELR3mut) (n = 6) | $P^{EM}_{vs\,Normal}$ | $P^{KDELR3mut}_{vs\,Normal}$ | $P^{KDELR3mut}_{vs\,EM}$ |
|---|---|---|---|---|---|---|
| Visual acuity | 5.03 | 5.00 | 4.80 | 0.302 | 0.077 | 0.116 |
| Refractive error (D) | -0.46 | -10.29 | -9.92 | 0.005 | 0.005 | 0.732 |
| Astigmatism | -0.46 | -2.33 | -2.34 | 0.088 | 0.005 | 1.000 |
| Axial length (mm) | 24.04 | 27.27 | 27.47 | 0.002 | 0.002 | 0.818 |
| IOP (mm Hg) | 10.23 | 14.52 | 15.73 | 0.045 | 0.020 | 0.297 |
| Subfoveal retinal thickness(um) | 246.5 | 237.7 | 282.8 | 0.103 | 0.762 | 0.480 |
| Subfoveal choroidal thickness(um) | 510.8 | 338.8 | 283.2 | 0.002 | 0.009 | 0.309 |

71.58% of the control group ($P = 7.58 \times 10^{-5}$, Wilcoxon rank sum test, Fig. 4e). To further determine the effect of *kdelr3* knockdown on visual function in zebrafish, we tested the VMR to evaluate visual behavior. Based on a previously described protocol[36], we applied three ON and three OFF light stimuli to 5 dpf larvae in 96-well plates. Morphants injected with 0.35 ng of *kdelr3*-MO displayed reduced activity in OFF

conditions ($P = 0.0088$, Wilcoxon rank sum test; Figs. 4f, g). Taken together, our results demonstrated that temporary silencing of the *kdelr3* gene led to significant severe visual impairments, consistent with the manifestations observed in EM patients. Our findings contribute valuable insights into the genetic mechanisms of EM predisposition driven by *KDELR3* mutations.

**Fig. 4 | Characterization of *kdelr3*-deficient human and zebrafish. a** SS-OCT images of right (OD) and left (OS) eyes for individuals with and without rare PTVs in *KDLER3*. **b** Quantification of eye measurement, retinal and choroidal thickness in normal eyes and EM eyes with *KDELR3*-deficient. *P*-value calculated by two-sided Wilcoxon rank sum test. **c** Quantification of eye axis length, the axis-to-body length ratio, lens diameter, and lens-to-body length ratio across various MO dose groups. Each group has more than five eyeballs. Data was analyzed by two-sided Student's *t*-test, ∗*P* < 0.05, ∗∗*P* < 0.01, ∗∗∗*P* < 0.001 significantly different from control 0.5 ng group (n.s. non-significant differences). Boxplots display the median, quartiles, and variability of the data. **d** Immunohistochemistry of control-MO injected eyes and *kdelr3*-deficient morphants. **e**, Boxplots representing the normalized fluorescence intensity associated with various MO dose groups. Each group has more than ten eyeballs. Data was analyzed by two-sided Student's *t*-test, *P* < 0.05, **P* < 0.01, ***P* < 0.001 significantly different from control group (n.s. non-significant differences). Boxplots display the median, quartiles, and variability of the data. **f** VMR testing in *kdelr3*-deficient zebrafish morphants. Larvae injected with *kdelr3*-MO (0.35 ng) showed a weaker ON response and a significantly attenuated OFF response compared to control larvae. **g** The data in the Figure represent averages ± standard deviation of triplicate assays in one experiment. ***P* < 0.01, ****P* < 0.001 (two-sided Student's *t*-test).

## RNA-seq analysis of downstream genes regulated by *KDELR3*

We then examined *KDELR3*-mediated downstream targets that could account for the thinning of choroidal and scleral thickness during the pathogenesis of EM. RNA-seq analysis was performed in the *kdelr3*-MO and the control zebrafish eyeballs, and *KDELR3*-shRNA RPE and HSF cells (Supplementary Fig. 16). Among the significantly differentially expressed genes (DEGs), transcripts of 1350, 956, 118 and 379 genes were upregulated, and transcripts of 277, 304, 69 and 1102 genes were downregulated in response to *KDELR3* knockdown compared to the controls (Fig. 5a, Supplementary Data 18–21). In addition, gene ontology (GO) enrichment analysis demonstrated significantly affected categories in genes that were downregulated or upregulated in response to *KDELR3* deficiency (Supplementary Data 22–25). The downregulated genes were associated with extracellular matrix organization (GO:0030198, R-HSA-1474244, R-DRE-3000178) and cell-cell junction organization (GO:0045216, 0034330) (Fig. 5b). Upregulated GO terms indicated neurotransmitter transport (GO: 0006836, 0046686, 0070588, 0030001) and blood vessel development (GO: 0001568, 1901342, 0002064) (Supplementary Fig. 17). We also observed a significant positive correlation in enrichment score for shared terms ($R = 0.92$, $P < 2.2 \times 10^{-16}$ and $R = 0.69$, $P < 2.2 \times 10^{-16}$; Fig. 5c) across zebrafish and cell types. Furthermore, the GSEA of the representative gene set in each *KDELR3* knockdown groups on extracellular matrix proteins is shown in Fig. 5d, which suggests that *KDELR3* is closely involved in ECM remodeling.

Among the 8160 significant DEGs regulated by *KDELR3* in RPE or HSF, over 27 ECM-related genes have previously been reported to be associated with myopia[37], including 18 significant downregulated genes (Fig. 5e). We then performed qRT-PCR to examine the representative 18 genes mediated by *KDELR3* (Fig. 5f). Consistent with the RNA-seq data, the results indicated that collagen (COL) family genes, including *COL1A1*, *COL4A1*, *COL4A3*, *COL4A5*, *COL5A1*, and *COL11A1*, were significantly downregulated after knockdown of *KDELR3* expression by shRNA in HSF cells and RPE cells ($P < 0.05$, Student's *t*-test; Fig. 5f, Supplementary Fig. 18). As we known, fibroblasts-to-myofibroblast transdifferentiation and the ECM alterations in sclera involved in myopia pathogenesis[18]. Furthermore, we used western blots to examine myofibroblast transdifferentiation and collagen production in HSF cells and RPE cells after knocking down the expression of *KDELR3*. In HSF and RPE cells, there was an increased expression of the myofibroblast marker α-SMA and a decreased expression of COL1α1 (Fig. 5g and Supplementary Fig. 19). Collectively, these data demonstrate that the ablation of *KDELR3* leads to scleral and choroidal thinning that results from ECM remodeling. This degeneration recapitulates the disease phenotype in the EM patients with *KDELR3* mutations.

## Discussion

Here we report, to our knowledge, the largest-scale WES study of EM conducted to date, and we provide a detailed characterization of its genetic landscape. We adopted ACMG standards[38] to classify variant pathogenicity and used uniformly processed data from a large control population to minimize false positives from possible subpopulations. Upon the 75 known causative genes[37,39], a total of 65P/LP variants were ultimately identified in 22.7% of the patients with EM. Despite this progress, the genetic defect of more than 75% of the patients with EM remains unknown as it has been challenging to screen all causative genes synchronously.

In this large EM exome study, we assembled the exomes of 449 EM cases and 9606 controls and observed an exome-wide enrichment of PTVs, which typically results in protein loss of function. Rare PTVs were found in genes involved in retrograde vesicle-mediated transport, which are significantly enriched in patients with EM. This reflects the highly polygenic genetic architecture of EM, a property shared with HM[31], and suggests that the majority of genes involved in EM risk will require larger sample sizes to be discovered. The existing cell-type enrichment methods mainly focus on common variants (e.g., LDSC-SEG, MAGMA-based scDRS, and RolyPoly), which cannot differentiate common and rare variants due to their limited statistical power for rare variants. Therefore, to analyze the burden of specific genes in ocular cells and to gain insights into the pathology and cellular origin of EM, we introduce scPBS, a method that incorporates rare quantified variants and scRNA-seq data to identify EM-relevant single cells. As with previous single-cell disease relevance score[24,40], we have demonstrated that rvTRS can associate individual cells with EM, assessing the heterogeneity across individual cells within predefined cell types in their association to EM, and broadly associating predefined cell types to EM. Single-cell gene expression data from the developing human embryonic eyes[23] implicate human scleral fibroblasts in EM risk.

For single gene discovery, we used a case-control collapsing analysis to identify three rare variants in eight patients with EM in the gene encoding the KDEL endoplasmic reticulum protein retention receptor *KDELR3*, accounting for ~1.78% of the population affected with EM. Additionally, in our MAGIC cohort of nearly 10,000 people with high myopia (single eye ≤ −6.00 D)[31], we found that patients with the *KDELR3* rare PTVs had worse refractive error than those without the mutation. Recently, Karczewski et al. determined a gene-based association study investigating 4529 phenotypes in 394,841 UK Biobank exomes (https://app.genebass.org/)[41]. We used the published UK Biobank portal to demonstrate phenome-wide association studies (PheWAS) of rare predicted loss of function (pLoF) in *KDELR3* on other diseases (Supplementary Fig. 20). There were 4528 pLoF gene burden associations with *KDELR3*, and *KDELR3* was most significantly associated with both corneal resistance factor left custom ($P = 4.04 \times 10^{-5}$) and corneal hysteresis left custom ($P = 4.23 \times 10^{-5}$). It is also known that high myopia has lower corneal hysteresis than emmetropes and low myopia[42]. The previous study found the KDELR-Gs-PKA signaling cascade regulates retrograde transport machinery[43]. The study indicated that the KDELR-Gs-PKA axis is a Golgi-based cell-autonomous device, or control system, that couples with components of the transport apparatus and uses both signaling and transcriptional networks to maintain homeostasis at the ER-Golgi interface. Disruptions in Golgi homeostasis can lead to abnormal ECM production and contribute to the development of various ECM-related disorders[44]. Therefore, the candidate gene, *KDELR3*, identified in this study is involved in several processes previously recognized as being involved in human myopia, such as ECM degradation[18] and endoplasmic reticulum (ER) stress[30] in sclera fibroblasts. The phenotypes associated with orthologous genes

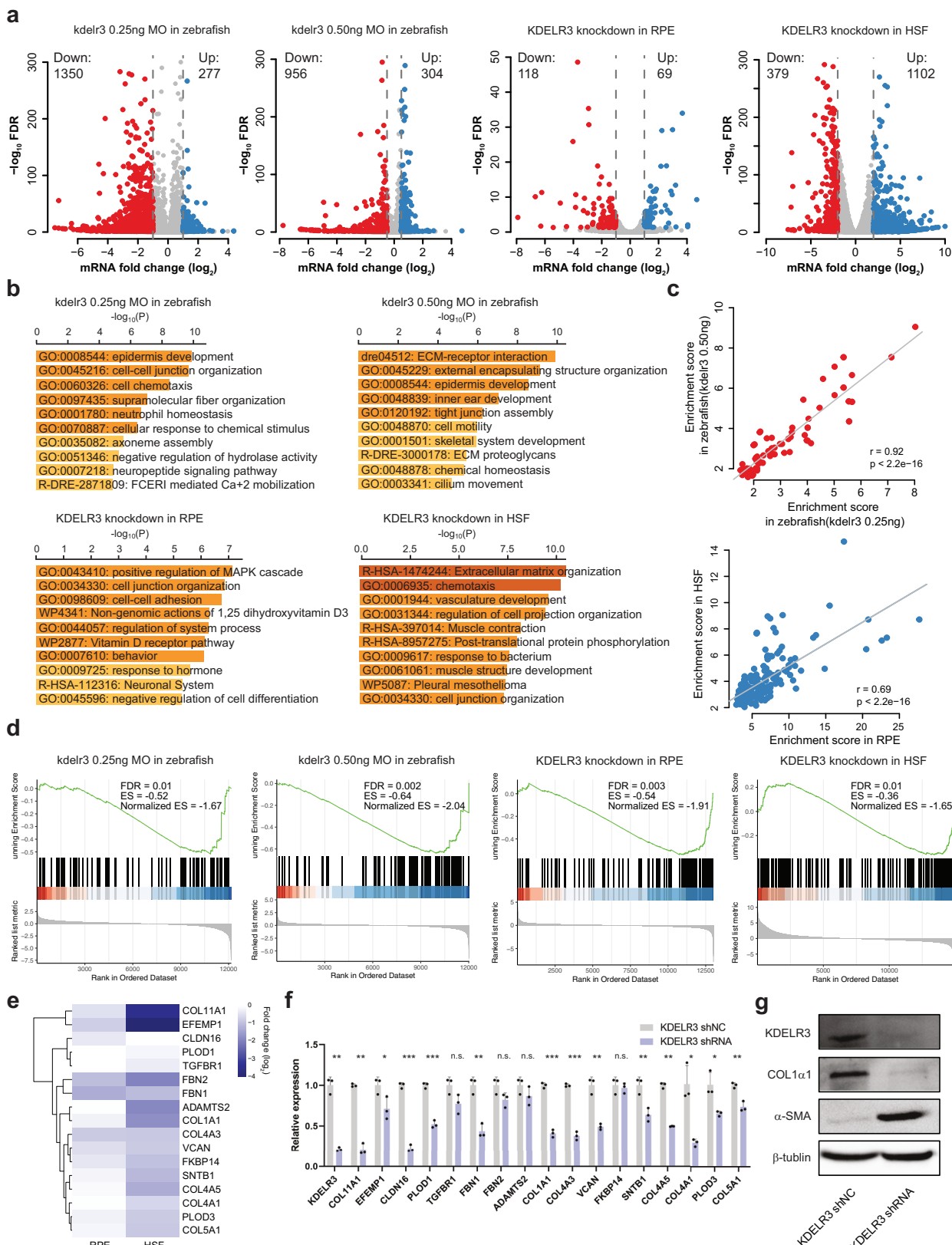

in corresponding zebrafish disease models support their roles in reduced collagen levels for ECM modeling and EM pathogenesis.

In the longer term, we speculate that WES of patients with EM, coupled with in-depth clinical phenotyping and image assessment, may improve the precision of classification schemes to prognosticate refraction outcomes and stratify patients for specific treatments (such as eyeglasses versus laser-assisted in situ keratomileusis procedure versus Implantable Collamer Lens versus pharmacological therapies)[45–47]. For example, in some forms of myopia with abnormal collagen (e.g., Marfan Syndrome and Stickler Syndrome), surgical LASIK procedure may merely expose patients with myopia to surgical morbidities, such as blindness, poor wound healing, poor

**Fig. 5 | RNA-seq analysis of *KDELR3*-modulated genes. a** A volcano plot illustrating differentially regulated gene expression from RNA-seq analysis between the control and *KDELR3*-deficient zebrafish eyeballs and human cell lines. Genes upregulated and downregulated are shown in red and blue, respectively. Values are presented as the log2 of fold change by DESeq2. **b** Top 10 Metascape clusters terms enriched for the downregulated targets in *KDELR3*-deficient transcriptome. The x-axis depicts -log (*P*-value). The p-value is defined as the probability of obtaining *n* or more pathway members, forming a cumulative hypergeometric distribution. **c** Gene ontology and pathway enrichment score are correlated for Metascape terms shared among downregulated genes of *KDELR3*-deficient zebrafish eyeballs and human cell lines. (Insert) Pearson correlation coefficients. **d** GSEA analysis for ECM organization in downregulated genes of *KDELR3*-deficient zebrafish eyeballs and human cell lines. normalized enrichment score (NES); false discovery rate-adjusted *P*-value (FDR). FDR derived from a GSEA analysis which uses a two-sided Kolmogorow–Smirnow test and performs multiple testing correction. **e** Heatmap of representative ECM organization-related genes. Each row represents the same gene in RPE and HSF cells. **f** qRT-PCR validation analysis shows the mRNA expression fold change of ECM organization-associated genes in the control vs *KDELR3*-deficient HSF cells. Each group has three biological replications. Data represent mean ± s.d. and *P*-value was generated by using two-sided Student's *t*-test. \**P* < 0.05; \*\**P* < 0.01; \*\*\**P* < 0.001. **g** Protein levels of scleral KDELR3, COL1α1, and α-SMA were detected by Western blot in *KDELR3*-deficient HSF. Source data are provided as a Source Data file.

refractive predictability, and globe rupture without addressing disease pathogenesis[48,49]. Surgical intervention in these contexts is unlikely to improve associated developmental phenotypes such as vitreous abnormalities, astigmatism, hypoplastic iris, or hypoplastic ciliary muscle, which are more likely to result from genetic defects in embryonic eye development than from reversible consequences of visual stimuli accumulation. These observations should raise thresholds for surgical intervention in patients with EM-disrupting type I collagen.

Due to the limitations of WES, systematic analyses of noncoding regions, copy number variations, and structural variations in EM were not conducted here. Insertion-deletions, rearrangements, noncoding variants, and intronic splice mutations are also likely to contribute to genetic risk for EM and will be subjects of our future studies. Beyond these technical limitations, our sample size in this cohort still lacks sufficient statistical power to detect many rare, sporadic EM-associated risk genes due to the high genetic heterogeneity of EM. One indication of this heterogeneity is the lack of significance in a large proportion of known myopia-causing genes. Despite the larger size of this study compared to previous efforts, the genetic contribution to EM reported here is likely to be an underestimation. Although our patients are mostly of European origin, international collaborative studies will soon overcome our current limitations of small cohort size and limited ethnic diversity.

In summary, this study provides a detailed characterization of potential pathogenic variants in EM, extending the genetic profile of EM. Larger cohort size, parent-proband trio sequencing, advanced genomic technologies, and international collaborative studies are critical to overcome the limitations of this study and to further elucidate the genetic etiologies of EM. In addition, mapping the genetic landscape of individuals with differences in common myopia, high myopia, and extreme myopia, may aid in understanding the common genetic factors in myopia progression.

## Methods
### Study design and participants
Our study included 449 individuals with EM at the Eye Hospital of Wenzhou Medical University (Zhejiang Eye Hospital, Wenzhou, China). The number of male and female patients is 226 and 223, respectively. EM was defined as an uncorrected visual acuity of 20/25 or less and a spherical equivalent refraction (SER) of −10.0 D or less (Supplementary Fig. 21). Individuals with EM were basically diagnosed by visual acuity and autorefraction testing[50]. Research performed on samples and data of human origin was conducted according to the protocols approved by the institutional review boards of the Eye Hospital of Wenzhou Medical University, and informed consents were obtained from all subjects, and were conducted in compliance with the Declaration of Helsinki. All procedures were carried out strictly following the guidelines of 'Management of Human Genetic Resources', as stipulated by the Ministry of Science and Technology of China (no. BF2022060511307 and no. BF2022060611309, effective from November 8, 2021).

### Whole-exome sequencing data generation
All EM samples were sequenced by Berry Genomics on Illumina NovaSeq 6000 sequencers with the use of the Twist Human Core Exome Kit. The FastQC package was used to assess the quality-score distribution of the sequencing reads. Read sequences were mapped to human genome build 37 (GRCh37)[51,52] using the Burrows-Wheeler Aligner (BWA 0.7.12)[53]. And Sambamba 0.6.6 (https://lomereiter.github.io/sambamba/) was used for sorting by chromosome coordinates and marking duplicates. After alignment by BWA, the reads were subjected to recalibration using the Genome Analysis Toolkit (GATK v. 4.0.11.0)[54]. Haplotype calling was performed by using HaplotypeCaller in GATK v.4.0 in GVCF mode according to the best practice. GVCFs were merged and joint genotyped with GenotypeGVCFs in GATK v.4.0 to produce a combined VCF file for further analysis. This pipeline detected SNVs and small insertion or deletion (indel) variants from exome sequence data.

### Quality control
We applied standard variant-level and individual-level quality controls. Variant calling accuracy was estimated using the GATK Variant Quality Score Recalibration (VQSR) approach. Then, we excluded variants for further analysis if (1) they were located inside of low-complexity regions; (2) they failed in GATK VQSR metric; (4) they had calling rates <90%; (5) Hardy–Weinberg Equilibrium (HWE) test *P*-value < $10^{-6}$ on the basis of the combined case and control cohort; (6) Genotypes with a genotype depth (DP) < 10 and genotype quality (GQ) < 20; and (7) heterozygous genotype calls with an allele balance >0.8 or <0.2. After that, we excluded samples with a low average call rate (<0.9), low mean sequencing depth (<10), or low mean genotype quality (<65). Outliers (>4 standard deviation [SD] from the mean) of the transition/transversion ratio, heterozygous/homozygous ratio, or insertion/deletion ratio within each cohort were further discarded (Supplementary Fig. 22). Samples with an X chromosome inbreeding coefficient >0.8 were classified as males, while samples with an X chromosome inbreeding coefficient <0.4 were classified as females. Samples between <0.8 and >0.4 which classified as ambiguous sex status, were excluded from the dataset (Supplementary Fig. 23). We detected population outliers and stratification using a method based on principal component analysis (PCAs) with a subset of high-confidence single-nucleotide polymorphisms (MAF >1%) in the exome capture region. Only retained individuals of East Asian (EAS) ancestry were retained, which were classified by a random forest algorithm with 1000 Genomes data (Supplementary Fig. 24). Within the EAS population, we down-sampling cohorts (449 cases and 449 controls) by removing the controls that were not well-matched with cases on the basis of the top three PCs by PCAmatchR[55] (Supplementary Fig. 25). We included only unrelated individuals (identity by descent proportion <0.2) using PLINK 1.07[56]. After QC, we retained 449 cases and 9606 controls.

### Interpretation of variants for known myopia genes
Annotation of pathogenic (P) or likely pathogenic (LP) variants was performed by using ANNOVAR[57], and pathogenicity was assigned

according to 2015 American College of Medical Genetics (ACMG) criteria using InterVar[58], which is a computational implementation of expert panel recommendations for clinical interpretation of genetic variants (ACMG 2015 criteria)[38]. ACMG classification categories include pathogenic (P), likely pathogenic (LP), variant of uncertain significance, likely benign, and benign. Variants that were rare (maximum population-specific minor allele frequency [MAF] < 1% in the Genome Aggregation Database [gnomAD][59], protein-altering (missense, splice site, stopgain, startloss, stoploss (none observed)), and classified as pathogenic or likely pathogenic by InterVar were retained for analysis. Human genes that were regarded as established causal genes for myopia had previously been identified with deleterious variants in individuals affected by either syndromic or isolated myopia. All myopia susceptibility genes were extracted from OMIM (Online Mendelian Inheritance in Man, https://omim.org/)[39], IMI-Myopia Genetics Report[37], and PubMed for articles published up to December 2022, using terms related to genes (for example, 'gene', 'genetic', 'mutation' or 'variant') in conjunction with terms related to myopia (for example, 'myopia', 'high myopia' and 'nearsightedness'). In total, a list of 75 known genes was compiled as well, and the associated phenotypes references for each known myopia gene are listed in Supplementary Data 3. Only variants classified as P or LP were reported here (Supplementary Data 4).

### Variant annotation

The annotation of variants was performed with Ensembl's Variant Effect Predictor (VEP v.99)[60] for human genome assembly GRCh37[61]. We used population AF from public databases: 1000 Genomes[62], ESP, and Genome Aggregation Database (gnomAD)[59]. We employed multiple in silico prediction algorithms including PolyPhen-2[63], SIFT[64], Combined Annotation Dependent Depletion (CADD)[65], LOFTEE[59], and SpliceAI[66] plugins to generate additional bioinformatic predictions of variant deleteriousness. Protein-coding variants were annotated into the following four classes: (1) synonymous; (2) benign missense; (3) damaging missense; and (4) PTVs. In detail, using VEP annotations (v.99), missense variants were classified as "inframe_deletion", "inframe_insertion", "missense_variant" or "stop_lost" variants. Among the missense variants, one type of benign missense (B-mis) was predicted as "tolerated" and "benign" by PolyPhen-2 and SIFT, respectively, and another type of benign mutation showed a combined annotation dependent depletion (CADD) score <15. Furthermore, damaging missense (D-mis) was predicted as "probably damaging" and "deleterious" by PolyPhen-2 and SIFT and CADD > 20. Finally, PTVs were classified as "frameshift_variant", "splice_acceptor_variant", "splice_donor_ variant", "stop_gained", or "start_lost" variants (Supplementary Data 6).

### Exome-wide single-variant association analysis

We estimated associations of common and low-frequency variants (MAF > 0.005) by using MLMA-LOCO[67] and EMMAX test[68] and correcting for sex and the first ten PCs. The test statistics obtained via linear regression were inflated because of the population differentiation caused by genetic drift[69]. Post hoc correction approaches, such as "Genomic Control", were used to correct the inflation[70].

### Gene-set burden analysis

To estimate the excess of rare, deleterious protein-coding variants in individuals with EM, we conducted burden tests across the entire exome and biologically relevant gene sets. We focused on definitions of rare genetic variants for the MAF less than 0.5% in our dataset, the 1000 Genomes, NHLBI Exome Sequencing Project (ESP), and gnomAD. We implemented gene-set burden tests by using Fisher's exact test (FET) and logistic regression to examine the enrichment of rare variants in individuals with EM versus controls. We collected 10,587 gene sets derived from the GO biological process ontology, KEGG,

REACTOME, and transcription factor targets from The Molecular Signatures Database (MSigDB)[22] and used a two-sided FET to assess whether the proportion of carriers among EM cases was significantly higher than among controls. For candidate gene sets, we also performed the logistic test by regressing case-control status on certain classes of rare variants aggregated across a target gene set in an individual and adjusting for sex, the top ten PCs, and exome-wide variant count.

### Gene-based collapsing analysis

For gene-based tests, we restricted our testing to deleterious rare variants annotated as PTVs. To assess whether a specific gene exhibited an overrepresentation or underrepresentation of rare PTVs in HM cases, we performed four gene-level association tests including Fisher's exact test, burden, SKAT, SKAT-O[71], and ACAT[72], with previously defined covariates (sample sex, PC1-PC10). The threshold of exome-wide correction for multiple testing was set as $P < 7.89 \times 10^{-06}$ (0.05/6335 genes), which was calculated as the 5% types I error rate divided by the number of genes. Instead of assuming a uniform distribution for $P$-values under the null, we generated empirical $P$-values by permuting case-control labels 1000 times, ordering the FET $P$-values of all genes for each permutation, and taking the average across all permutations to form a rank-ordered estimate of the expected $P$-value distribution.

### Cell-type enrichment

We acquired the single-cell RNA-seq (scRNA-seq) expression matrix of developing human embryonic eyes from NCBI's GEO with the GEO Series identifier GSE228370 and performed a scRNA-seq analysis following the methodology outlined in the original publication[23]. The Seurat standard procedure (v4.3.0) was used for quality control, dimensionality reduction, clustering, filtering, and doublet exclusion[73]. Quality control was performed in multiple steps. Initially, genes expressed in less than 3 cells, and cells with less than 200 transcripts were removed from the analysis. Then we filtered out cells with ≤200 or ≥8000 expressed genes and ≥20% mitochondrial genes as described in their study. The gene expression matrix was normalized and scaled using the NormalizeData and ScaleData functions. The top 2000 variable genes were selected using FindVariableFeatures for PCA analysis. The batch effect among samples was removed by Harmony (v1.0) using the top 20 principal components from PCA. Cells were divided into 69 clusters using FindClusters with a resolution parameter at 2.5 and the top 20 principal components. Cell clusters were visualized using UMAP with Seurat function RunUMAP.

We obtained the marker genes of each predefined cell type compared to all other cell types in the dataset using Seurat's FindAllMarkers function[73] with the following parameters: only.pos = TRUE, min.pct = 0.1, logfc.threshold = 0.25, test.use = "MAST", assay = "RNA". The 10% most specific genes in each cell type were tested for rare PTVs enrichment in EM cases.

### scPBS methodology

Gene expression was scaled to a total of 1 million UMIs (or transcripts per million (TPM)) for each cell type. We then calculated a metric of gene expression specificity by dividing the expression of each gene in each cell type by the highest expression of that gene in all cell types, leading to values ranging from 0 to 1 for each gene. Quantified rare PTVs are used to build the variant-by-individual matrix of EM derived from the MAGIC pipeline (https://github.com/sulab-wmu/MAGIC-PIPELINE). The scPBS algorithm is described below and Python software is available at GitHub (https://github.com/sulab-wmu/MAGIC). Given an EM quantify variant matrix and a scRNA-seq data set from human eyes, scPBS computed a rare-variant-based trait-relevant score (rvTRS) for each individual cell of association with the EM.

The framework of the method was described in the following steps. First, scPBS assembled a collection of highly expressed genes

that are specific to each cell. These genes were determined based on their high cell-to-cell variability within the dataset, identifying the most specific genes in each individual cell. Second, scPBS selected rare PTVs that passed all filters of a specific model and built the gene set(cell)-by-individual indicator matrix used for the polygenic burden test. Third, scPBS quantified EM-related polygenic burden coefficient (odds ratio) in each cell to generate single-cell-specific polygenic burden scores (scPBS). Following previous studies[40], we correlated scPBS with the expression level of each gene across cells and prioritized the trait-relevant genes by ranking the Pearson correlation coefficients (PCCs). Finally, a rare-variant-based trait-relevant score (rvTRS) of each cell computed by averaging the expression level of the top 10% trait-relevant genes based on ranked PCCs and subtracting the random control cell score via the AddModuleScore cell-scoring method used in Seurat with default parameters[73].

### Single-cell RNA-seq analysis for candidate gene

scRNA-seq profiles from the adult choroid[34] and sclera[33] were used to identify cell-type specificity of newly identified candidate genes. scRNA-seq expression values were log-transformed and centered using the mean expression values. The average centered expression values of candidate genes were calculated for each cell. Cells were grouped into cell clusters (fibroblasts, choroid endothelial cells, veins, arteries, etc.), and the relative expression level of a given cell cluster was calculated by a scale function in R.

### Measurement of morphologic parameters using optical coherence tomography image

All the patients with rare PTVs in *KDELR3* and controls had dilated funduscopic examinations. Best-corrected vision acuity (BCVA), slit-lamp biomicroscopy, AL (IOL Master; Carl Zeiss Meditec, Dublin, CA), spherical equivalent (SE), and intraocular pressure (IOP) were measured. The visual acuity was measured with a standard logarithmic visual acuity chart, and the decimal visual acuity was converted to logMAR units for statistical analyses. OCT imaging of the subfoveal retina area was obtained using the OCT device (ZEISS CIRRUS HD-OCT 5000) by an experienced ophthalmologist. For the measurement of the subfoveal choroidal area, a swept-source optical coherence tomography (SS-OCT, ZEISS PLEX Elite 9000) was chosen. The choroidal layer was automatically segmented using the proprietary algorithm from the outer edge of the hyper-reflective retinal pigment epithelial line to the inner edge of the sclera.

### Transient suppression of *kdelr3* and in vivo complementation studies in zebrafish

Husbandry adult zebrafish of the Tg(kdrl:mCherry), Tg(gad1b:Cherry), and Tg(gfap-eGFP) strains were obtained from the China Zebrafish Resource Center (CZRC Catalog ID CZ921). All experiments were carried out in accordance with the Association for Research on Vision and Ophthalmology's statement on the Use of Animals in Ophthalmic and Vision Research and were approved by the Institutional Animal Care and Use Committee of Wenzhou Medical University. Adult zebrafish in 3-month-old ($n = 2$ for each PCR sample) were used for tissue-specific qRT-PCR. Zebrafish embryos ($n = 20$ for each PCR sample) at different time points from 1-day post fertilization (dpf) to 7 dpf were harvested for time series qRT-PCR. Total RNA was extracted with Trizol, and the cDNA products were used for qRT-PCR with SYBR green (Roche Applied Science, Germany). Primer sequences are listed in Supplementary Data 26. The relative gene expression was quantified with an ABI Q6 detection system.

Reciprocal blast of human *KDELR3* (GenBank: NP_057839.1) against the D. rerio genome identified a single zebrafish ortholog (77% amino acid identity). We designed a splice-blocking morpholino (MO) targeting the splice donor site of exon 2 (5′-TCCACAGAATCTACACA-CACACACA-3′; Gene Tools). The *kdelr3* targeting MO was microinjected into the yolks of one-cell stage embryos with four different doses (0.15, 0.25, 0.35, and 0.5 ng). Morphological analyses were performed at 3dpf[74,75]. To validate the splice modifying effect of the *kdelr3* targeting MO, RT-PCR was performed with the primers listed in Supplementary Data 27. RT-PCR products were analyzed by electrophoresis analysis (Supplementary Fig. 13) and Sanger sequencing.

### *Kdelr3* mRNA preparation and rescue experiments

To reverse the observed knockdown effect in *kdelr3*-deficient morphants, we introduced the entire zebrafish *kdelr3* cDNA into the pUC57 vector, incorporating the T7 promoter sequence and Kozak sequence at the 5′ end. The cDNA templates were directly synthesized by Sangon Biotech (Sangon, China). For amplification, we employed primers containing the T7 promoter for zebrafish *kdelr3* cDNA: 5′-TAA-TACGACTCACTATAGGGGCCACCATGAACATCTTCCG-3′ and 5′- TCA-CACCGGCATTGGCAGAGACATCTTTCC-3′. Following high-fidelity PCR, the template DNA was purified using the QIAquick PCR Purification Kit (Qiagen, Germany). Capped full-length mRNAs were synthesized using the mMESSAGE mMACHINE T7 ULTRA Transcription Kit (Invitrogen, United States), and subsequently purified with an RNeasy Mini Kit (Qiagen) according to the manufacturer's instructions. Then, full-length zebrafish *kdelr3* mRNAs were co-injected with a targeting MO into one-cell stage embryos at a final concentration of 200 ng/μL

### Ocular measurement and morphological analysis

For data collection, 8–15 larvae were included in each trial, and the trials were performed in triplicate. As previously described[74], in vivo imaging of the lateral and vertical view of each larva was captured by a stereoscopic fluorescence microscope (SZX16, OLYMPUS, Japan). Body length, axial length, and lens diameter were calculated by the built-in program (OLYMPUS cellsens standard 1.14).

### Visual motor response

Visual motor response (VMR) was performed in a ZebraBox (VMR machine ViewPoint 2.0, France) in order to assess the larvae's response to light at 5 dpf[75]. For data collection, each experimental group included 12 larvae in a 96-well plate, and 3 h of dark adaption was allowed before the behavior tests. A ZebraBox (VMR machine; ViewPoint 2.0, France) was set to give 3 rounds of ON and OFF light stimuli (30 min for each stimulus) to the larvae. The larval activity during the 150 s spanning the time of light switching was recorded.

### Immunohistochemistry

Zebrafish morphants were fixed in 4% paraformaldehyde at 5 dpf and gradually dehydrated with 15, 22.5, and 30% sucrose. Subsequently, 10 μm-thick frozen sections were collected and stained with recoverin (RCVRN) antibody (rabbit; 1:1000; proteintech), respectively at 4 °C overnight. The sections were incubated with donkey anti-mouse IgG secondary antibodies conjugated with Alexa Fluor 594 (1:200) for 2 h at room temperature. DAPI (4,6-diamidino-2-phenylindole) was used for nuclear staining. After the coverslips were mounted, images were captured with a DM4B microscope (Zeiss, Oberkochen, Germany).

### Cell cultures and lentivirus infection

The human cell lines RPE and human scleral fibroblasts (HSF) were provided by Dr. Yutaka Shimada (Kyoto University, Kyoto, Japan) and Dr. Xiangtian Zhou (Wenzhou Medical University, Wenzhou, China), respectively. The RPE cells were cultured in RPMI-1640 medium (RNBL4082, Sigma) with 10% FBS with 100 U/ml of penicillin and 100 U/ml of streptomycin (15140-122, Gibco). HSF cells were cultured in DMEM (Dulbecco's Modified Eagle Medium) medium (12430054, Gibco) with 10% FBS, 1% Pre Strep, and 1% GlutaMAX Supplement (35050061, Gibco). HEK-293 cells were obtained from ATCC (CAT#CRL-1573), maintained in DMEM, with 100 U/ml of penicillin, 100 U/ml of streptomycin, and 10% heat-inactivated fetal bovine

serum. All cells were cultured in a cell incubator with 5% $CO_2$ at 37 °C. All of the cells were authenticated by short tandem repeat (STR) analysis and regularly tested for mycoplasma contamination.

## Plasmids design and construct

The shRNA plasmids were constructed by restriction cloning downstream of the U6 promoter using AgeI and EcoRI cut sites on pLKO.1 - TRC cloning vector (Addgene # 10878) by standard protocol. For a typical reaction, DNA oligo was diluted to 100 μM with ddH$_2$O and annealed in NEB 2.1 buffer, and heated at 95 °C for 2 min before slowly cooling to room temperature. Ligation was carried out by adding 1 μL T4 DNA ligase (5 U/μL, Thermo, EL0011) to give a total ligation reaction volume of 10 μL.

## Lentiviral package and transfection protocol

For lentivirus packaging, HEK-293 cells were seeded at $2.5 \times 10^6$ cells per dish on 10 cm dishes in DMEM medium. 24 h after seeding, 5 μg psPAX2 plasmid (Addgene #12260), 2.5 μg pMD2.G plasmid (Addgene #12259), and 5 μg shRNA plasmid were mixed and prepared with transfection reagent (Yeason, 40802ES08) following the recommended protocol from the vendor. 72 h after transfecting, lentivirus was collected following the recommended protocol, concentrated overnight using a Universal Virus Concentration Kit (Beyotime, C2901S), and used within 1–2 days to transduce RPE or HSF cells without a freeze-thaw cycle.

RPE or HSF cells were infected with lentivirus in a medium supplemented with 8 mg/mL polybrene (Sigma-Aldrich) for 48 h and were then selected with 1 mg/mL puromycin (60210ES25, Yeason) for two weeks. shRNAs used in this study are listed in Supplementary Data 28.

## RNA extraction, cDNA preparation, and quantitative real-time PCR

The total RNA was extracted using FastPure Cell/Tissue Total RNA Isolation Kit V2 (Vazyme Biotech Co., Ltd, RC112) according to the manufacturer's instructions. The RNAs were reverse-transcribed using HiScript III RT SuperMix for qPCR (+gDNA wiper) (Vazyme, R323) to synthesize cDNA. qRT-PCR was performed on a CFX Manager Real-Time PCR system (Bio-Rad CFX Manager) using specific primers and Taq Pro Universal SYBR qPCR Master Mix (Vazyme, Q712). The results were calculated as ΔΔCT using GAPDH as an internal reference transcript. ΔΔCT was calculated as ΔCT of the Knock down−ΔCT of the control. To work out the fold of gene expression, we performed 2-ΔΔCT. The primer sequences were provided in Supplementary Data 28.

## Western blot analysis

HSFs and RPEs were homogenized in RadioImmunoPrecipitation Assay buffer (RIPA Lysis Buffer, Beyotime, Shanghai, China) supplemented with a protease inhibitor cocktail (Roche, Grenzach Wyhlen, Germany) and 1 mM phenylmethanesulfonyl fluoride (PMSF, Beyotime). Total protein was extracted and the concentration was determined using an Enhanced BCA Protein Assay Kit (Beyotime). Equal amounts (30 μg) of total protein from HSFs or RPEs, were loaded onto a 10% sodium dodecyl sulfate-polyacrylamide gel, separated by electrophoresis, and then electro-transferred onto a nitrocellulose membrane (Millipore, Billerica, MA, USA). After blocking with 5% non-fat milk for 1 h at room temperature, the membranes were incubated with primary antibodies overnight at 4 °C. Primary antibodies against KDELR3 (proteintech, 27632-1-AP, 1:500), COL1A1 (proteintech, 67288-1-Ig,1:5000), a-SMA (Abcam, ab5694, 1:500), β-tublin (proteintech, 10094-1-AP, 1:5000) were used. After washing three times for 10 min each with tris-buffered saline containing Tween detergent (TBST, 10 mM Tris-HCl, pH 7.2–7.4, 150 mM NaCl, and 0.1% Tween-20), the membranes were incubated for 1 h at room temperature with goat anti-rabbit IgG (Beyotime, A0208, 1:1000) or goat anti-mouse IgG (Beyotime, A0206, 1:1000) antibodies.

The membranes were washed again three times in TBST, followed by visualization of protein bands using the Odyssey Infrared Imaging System (LI-COR Biosciences). Densitometric analysis of protein bands was performed using ImageJ software (Version 1.48v, National Institutes of Health [NIH], Bethesda, USA; https://imagej.nih.gov/ij). Values were normalized to those of the β-tubulin loading control.

## RNA-seq and bioinformatics analysis

Ten pairs of samples (eyeballs 5dpf zebrafish, including six mutant and six wild-type, and Cells from WT/MUT RPE and HSF cells) were collected. The total RNA was extracted using FastPure Cell/Tissue Total RNA Isolation Kit V2 (Vazyme Biotech Co., Ltd, RC112). Complementary DNA libraries were constructed using an Illumina TruSeq RNA Sample Prep kit according to the manufacturer's protocol. A total of 150 base paired-end reads were sequenced using the Illumina HiSeq X-Ten platform. RNA-seq raw reads were mapped to the human genome version hg38 using STAR-2.6.1[76] with default parameter values, used the htseq-count function in HTSeq[77] to calculate read counts for each gene. Differential gene expression analysis between groups was performed with the DESeq2 R package[78,79]. We used Metascape[80] for pathway analysis. Each pathway with a P-value (accumulative hypergeometric distribution) smaller than 0.01, and an enrichment factor >1.5 (the enrichment factor is the ratio between the observed counts and the counts expected by chance). Gene set enrichment analysis (GSEA) was conducted through the Bioconductor package 'clusterProfiler'[81]. Estimation of kdelr3-MO-induced splicing defect was identified using SUPPA2[35]. SUPPA2 extracts the PSI value for each event by transformation of alignment-free transcript quantification. Statistical significance is then calculated using a Kolmogorov–Smirnov test to compare the PSI distributions of two conditions as a function of the expression of the transcripts defining the events.

## Statistical analysis

The statistical analyses were performed using R software (version 3.5.1). Statistical significance was determined using the Student's t-test or Wilcoxon rank sum test, Linear regression analysis, or the one-way ANOVA with Bonferroni correction (more than two groups). Statistical significance was defined as a P-value less than 0.05. *P < 0.05, **P < 0.01, ***P < 0.001.

## Reporting summary

Further information on research design is available in the Nature Portfolio Reporting Summary linked to this article.

# Data availability

The raw genetic sequencing data for EM patients and control individuals generated in this study have been deposited in the Genome Sequence Archive for humans (GSA, https://ngdc.cncb.ac.cn/gsa-human/) under accession numbers HRA007816 in BIG Data Center, Beijing Institute of Genomics (BIG), Chinese Academy of Sciences. Raw RNA-seq data generated in this study have been deposited in the GSA for humans under the accession number HRA005720. All raw sequencing data deposited in GSA are under restricted access and will be approved via email to Jianzhong Su (sujz@wmu.edu.cn). A response would be expected within a week. Single-cell RNA-seq (scRNA-seq) expression matrix of developing human embryonic eyes from GSE228370. Source data for this paper are provided at with this paper in the Figshare repository at https://doi.org/10.6084/m9.figshare.24282832.

# Code availability

All the code used is publicly available at https://github.com/sulab-wmu/MAGIC-PIPELINE and https://github.com/sulab-wmu/MAGIC. Variant calling was performed using Genome Analysis Toolkit (GATK) (https://software.broadinstitute.org/gatk/). Quality control of

individual-level data was performed using PLINK v.2.0.a (https://www.cog-genomics.org/plink/2.0/). Variant annotation was performed using VEP v.95 (https://github.com/Ensembl/ensembl-vep) with the LOFTEE plugin (https://github.com/konradjk/loftee) and SpliceAI plugin (https://github.com/Ensembl/VEP_plugins/blob/release/107/SpliceAI.pm).

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

## Acknowledgements

We thank Dr. Jian Yang for advice on statistical analysis. We thank Dr. Zhenhui Chen and Dr. Yunlong Ma for their constructive comments regarding this manuscript. We thank the Berry Genomics Co., Ltd and OE Biotech. Co., Ltd (Shanghai, China) for sequencing services. This work was supported by the National Natural Science Foundation of China (81830027, U20A20364) and the Zhejiang Provincial Key Research and Development Program Grant (2021C03102) to J.Q.; the National Natural Science Foundation of China (82172882) to J.S.

## Author contributions

The study was conceived, designed, and supervised by J.S., J.Q., and J.Y. Analysis of data was performed by J.Y., Yue Z., K.L., J.L., Z.C., Y.Y., and Xiangyi Y. The experiments were conducted by You-Yuan Z., X.L., D.L., R.Z., F.Z., X.Z., and H.C. Patient sample recruitment was conducted by members of Myopia Associated Genetics and Intervention Consortium. DNA extraction and sequencing were carried out by Xiaoguang Y. The manuscript was written by J.Y. and J.S. with contributions from all other authors.

## Competing interests

The authors declare no competing interests.

## Additional information

## Myopia Associated Genetics and Intervention Consortium

Jianzhong Su [1,2,3] ✉, Liangde Xu[1], Jia Qu[1,2], Fan Lyu[1,2], Jian Yang[3], Hong Wang[1], Xiaoguang Yu[5] ✉, Jian Yuan[1], Yinghao Yao[2], Zhen Ji Chen[1], Yunlong Ma[6], Zhengbo Xue[1], Hui Liu[1], Wei Dai[1], Riyan Zhang[1] & Ran Zhuo [1]

[6]Department of Psychiatry, Perelman School of Medicine, University of Pennsylvania, Philadelphia, PA, USA.

