## [Peer Review File · Nature Communications]

Exome-wide Association Study Identifies KDELR3 Mutations in Extreme MyopiaREVIEWER COMMENTS

Reviewer #1 (Remarks to the Author):

Yuan et al described an interesting study to promote KDELR3 gene in association with extreme myopia (EM). In a large cohort (N=449) of individuals with EM, the authors identified diagnostic variant in 22% of cases with syndromic or nonsyndromic disorders. The authors then used case-control based exome sequencing analysis to identify PTV enrichment, especially in genes within retrograde vesicle-mediated transport, Golgi to ER pathways. Single cell RNA seq analysis identified fibroblasts in EM etiology. Gene based association study using case-control data identified several genes associated with EM, namely KDELR3 with strongest signal from the enrichment study, and high expression in fibroblast from bulk RNA seq study. MO knockdown zebrafish model was further introduced to demonstrate the phenotypic and biological impact of KDELR3 dosage to EM etiology.

The study design seems logical and the data are well presented. However, the reviewer strongly suggests to include a statistics expert to thoroughly review the stats of this paper. The current review is majorly focused on clinical and biological aspects. The reviewer was not able to see supplemental tables.

Major points

1. Description of the case and control cohorts seem to be lacking. What is the comparison regarding the ancestry, age, sex, etc between cases and controls? How do the cases match with the controls? Was the analysis controlled with variances such as age?
2. The reviewer cannot access supplemental table 5 to review P/LP diagnostic variants. However, according to the method description, the variant classification is solely based on InterVar. It is well known that InterVar classification is error prone. All variant classification need to be reviewed by field expert to ensure accuracy.
3. Any overlap between cases with KDELR3 PTV and ones with diagnostic variants? Are the cases with same KDELR3 variants unrelated?
4. How is KDELR3 related to retrograde vesicle mediated transport and pathway that is also found to be enriched with PTVs? The link is missing, and the mechanism needs to be better explained and linked.

Minor points

1. Line 232: This is a comparison between three EM cases with KDELR3 PTV versus three controls. However, the evidence do not support the conclusion. It is obviously there are clinical differences by comparing EM cases to controls. It is difficult to understand how to conclude that KDELR3 plays a role in such difference based on this comparison. If you randomly chose three EM cases regardless of KDELR3 status, would you expect a similar or different clinical difference?
2. Line 243: when increasing MO dosage, the phenotype seems to be different or reversed. Any proposed mechanism?
3. Line 344: it is premature to call KDELR3 PTV a "pathogenic variant". The gene-disease relationship and mechanism are not definitive at this time, because the current evidence are mostly from case-control signals and zebrafish model of gene knockdown. The findings are also not replicated in independent cohorts.

Reviewer #2 (Remarks to the Author):

Thank you for the opportunity to review the manuscript authored by Yuan et al., which offers a comprehensive WES genetic analysis concerning extreme myopia (EM). This study not only identifies new genes, particularly KDELR3, but also delineates relevant cell types associated with EM. Moreover, it includes a functional evaluation of KDELR3 within a zebrafish model.

While the manuscript adeptly presents these findings, there are issues that warrant attention.

Key aspects for consideration include:

WES analysis:

The authors included 449 cases and 9,606 controls in their EM WES analysis, with no evidence linking common and low-frequency variants with EM. Given the relatively small sample size, this lack of findings is understandable. Subsequently, a gene-based rare variant analysis was conducted, identifying KDELR3 and VN1R4 as significant. In light of the binary classification of EM, it would be beneficial to contextualize these results within the broader spectrum of myopia, refraction error, and high myopia, incorporating both previous findings by the authors and those by other groups.

Cell type enrichment:

The authors employed two methods to associate scRNA-seq with the polygenic burden of rare variants: a cell type level test, which showed no significant enrichment, and a single-cell resolution polygenic burden score (MAGIC, fibroblasts). Considering existing frameworks for cell type enrichment tests (e.g., magma-based, LDSC-based, scDRS), the uniqueness and advantages of the approach used in this study are not clear.

Reviewer #3 (Remarks to the Author):

The manuscript by Yuan et. al. describes the identification of endoplasmic reticulum protein retention receptor 3, KDELR3, as being a novel gene involved in the progression of extreme myopia (EM) whereby spherical equivalent refractive error (SER) is characterised as < -10.00 D. Determining the causative factors involved in the progression to EM is imperative in curbing the increasing health and socio-economic burden resulting from vision loss. The authors have shown compelling and thorough evidence that identifies KDELR3 rare protein truncating variants (PTVs) in patients suffering EM by using a range of techniques such as whole-exome sequencing (WES), single-cell RNA-sequencing (scRNA-seq), and bulk-cell RNA-sequencing (bcRNA-seq). This study postulates that KDELR3 expressed in scleral fibroblast are responsible for scleral extracellular matrix (ECM) organization and the regulation of both scleral thickness and strength. The authors have also investigated the function of KDELR3 in zebrafish larvae by generating the first transient knockdown (KD) model of KDELR3 using slice-modifying morpholino (MO) anti-sense oligonucleotides. The transient KD of KDELR3 resulted in an eye phenotype consistent with myopia, increased axial length and lens diameter. The authors have conducted extensive genetic and molecular studies to investigate the modulation of expression in genes downstream of KDELR3.

This original work provides a framework for further investigation into the molecular signalling of KDELR3 and the role it plays within ECM organization and the modulation of scleral thickness in EM. As highlighted by the authors, there is considerable evidence to suggest genetics primarily drive the succession of extreme myopia and as such it is unlikely increased environmental risk factors such as near-work are major contributing factors. As a result, it remains imperative to conduct large-scale population genetic screening to further investigate the genetic origin of EM. The work conducted by Yuan et. al. is coherent and supports the conclusions of the study and has contributed to the catalogue of known EM-associated genes.

The methodology and statistical analysis are sound and meets the expected standard in the field and is detailed enough to be reproduced. Some questions relating to the zebrafish MO model are listed below:

-As the authors are the first to describe a zebrafish KDELR3 splice modifying MO model, can they confirm KDELR3 KD up to 5 dpf using PCR and sequencing to identify any inclusion of introns or

frameshifts? The target exon is essential in generating splice modifying MOs. Is it possible to include this in supplementary figures as it is important to validate new MOs.

-Optimal dose testing off target effects typically range from 1.5ng – 6ng, with 5ng or less typically being the most effective and exhibiting the most biologically specific phenotype. The doses used by the authors are much lower than this range, using 0.15 – 0.5 ng, can the authors comment on why such low doses were used? Calibration of delivery in each experiment is critical.

-ZFIN currently has not described expression of KDELR3 in the visual systems. Perhaps you could update this atlas.

-Can the authors use SS-OCT imaging to calculate axial length and lens diameter in the MO model, normalised to body length?

Minor comments:

-Figure titles missing under the primary figures.

-Do the authors plan to generate a KO line to further investigate the role of KDELR3?

-A clearer Western blot showing decreased COL1a1 protein levels in the KDELR3 shRNA sample would be more convincing as the one shown suggests a problem with transfer.

-Supplementary Figure 13 is named Supplementary Fig. 1

-Supplementary Figure 14 is named Supplementary Fig. 2

Point-to-point response to the reviewers (NCOMMS-23-50564-T)

Reviewer #1

Comments for the Author

Yuan et al described an interesting study to promote KDELR3 gene in association with extreme myopia (EM). In a large cohort (N=449) of individuals with EM, the authors identified diagnostic variant in 22% of cases with syndromic or nonsyndromic disorders. The authors then used case-control based exome sequencing analysis to identify PTV enrichment, especially in genes within retrograde vesicle-mediated transport, Golgi to ER pathways. Single cell RNA seq analysis identified fibroblasts in EM etiology. Gene based association study using case-control data identified several genes associated with EM, namely KDELR3 with strongest signal from the enrichment study, and high expression in fibroblast from bulk RNA seq study. MO knockdown zebrafish model was further introduced to demonstrate the phenotypic and biological impact of KDELR3 dosage to EM etiology.

The study design seems logical and the data are well presented. However, the reviewer strongly suggests to include a statistics expert to thoroughly review the stats of this paper. The current review is majorly focused on clinical and biological aspects. The reviewer was not able to see supplemental tables.

Reply - We greatly appreciate reviewer #1's recognition of the quality and importance of our work. We have invited a statistics expert, Dr. Jian Yang, a professor of statistical genetics from Westlake University, to reviewed the statical analysis of the manuscript. In addition, we apologize that the Supplementary Tables were not included in the material available for review. Actually, the Supplementary Tables were provided separately as an Excel file in the submission. We sincerely hope that you can access the Supplementary Tables from the revised manuscript in resubmission.

Major points

1. Description of the case and control cohorts seem to be lacking. What is the comparison regarding the ancestry, age, sex, etc between cases and controls? How do the cases match with the controls? Was the analysis controlled with variances such as age?

Reply - Thank you for your valuable comments. We have added Supplementary Figures 22-25 to describe case and control cohorts at each QC step. A series of principal component analyses (PCAs) were performed to identify ancestral backgrounds and control for population stratification. Only individuals of East Asian (EAS) ancestry were retained, and were classified by a random forest algorithm with 1000 Genomes data (Supplementary Figure1, Supplementary Table1). Within the EAS population, we removed the controls that were not well matched with cases based on the top three PCs

by PCAmatchR. The ExWAS analysis included covariates of PCs of genetic ancestry (those among the first 10 PCs), sample sex and age.

Supplementary Fig. 22. Initial sample quality control analysis. Distribution of sample call rate, sample mean depth, sample mean genotype quality, sample transition to transversion ratio, sample heterozygous to homozygous ratio and sample insertion to deletion ratio. The box and whisker plots display the mean, minimum, and maximum.

Supplementary Fig. 23. Distribution of the inbreeding coefficient for 449 EM cases (a) and 9606 controls (b). Samples with an X chromosome inbreeding coefficient > 0.8 were classified as males, while samples with an X chromosome inbreeding coefficient < 0.4 were classified as females.

Samples with an X chromosome inbreeding coefficient between <0.8 and >0.4 which were classified as ambiguous sex status, and were excluded from the dataset.

Supplementary Fig. 24. Principal component analysis with 1000 Genomes. PCA was run on the study samples along with 1000 Genomes (1KG) phase 3 super populations to infer genetic ancestry. Genetic ancestry of the EM cases and controls were predicted by the Random Forest classifier based on top 6 PCs, using 1KG samples as the training data. Individuals were assigned to a particular 1KG-ancestry with a predicted probability > 0.9 , as depicted in the figure.

Supplementary Fig. 25. PCA on EM cases and controls samples. PCA after down-sampling by removing controls which did not pair-matched with cases based on the top 3 PCs.

2. The reviewer cannot access supplemental table 5 to review P/LP diagnostic variants. However, according to the method description, the variant classification is solely based on InterVar. It is well known that InterVar classification is error prone. All variant classification need to be reviewed by field expert to ensure accuracy.

Reply – Many thanks for the suggestions. We have invited a human genetic expert to ensure the accuracy of variant classification. We have added variant annotation in known myopia genes, which was evaluated by manual review according to the guidelines of the American College of Medical

Genetics and Genomics (ACMG) in Supplementary Table 4. According to the variant classification, we have updated the diagnostic mutations for EM cases in the Results section.

Supplementary Fig. 1. Overview of mutations identified in known myopia genes. a, Distribution of pathogenic (P), likely pathogenic (LP), uncertain significance (VUS), likely benign (LB) and benign (B) variants across all myopia known genes. b, Distribution of PTVs, missense, synonymous variants and other types (including in-frame indels) of all detected variants across genes. c, Distribution of PTVs, missense, synonymous variants and other types of P/LP variants across genes.

Figure 1. Study flowchart and overview of P/LP variants in known EM genes.

a, Flow chart for the study design to characterize and identify deleterious variants in known and novel EM gene. A total of 449 EM patients were recruited and underwent visual acuity examination, autorefractometry, and whole-exome sequencing. b, Allele counts of P/LP variants detected in 21 of 75 known EM genes. c, Diagnostic yield of known EM-causing genes in 449 patients. d, The proportion of each mode of inheritance in the 41 patients carrying P/LP variants in known EM genes. e, The proportional contribution of each gene among 41 cases. f, The proportion of patients classified according to the clinical syndromic information of the affected genes. g, GO pathway enrichment for 21 known EM genes. This network shows terms with a P -value < 0.01 , a minimum gene count of 3, and an enrichment factor > 1.5 . The nodes sizes are scaled with P -value. The known EM genes are most significantly enriched in visual perception, eye development and extracellular matrix organization. The number in parentheses shows the gene number and proportional contribution of each gene. h, Schematic overview of proportional contribution of retinal cell-specific genes.

3. Any overlap between cases with *KDEL3* PTV and ones with diagnostic variants? Are the cases with same *KDEL3* variants unrelated?

Reply - Thank you for your important comments. There was no case that both had rare PTV in *KDELR3* and P/LP variants in 75 candidate myopia genes (Supplementary Fig. 9).

Supplementary Fig. 9. Overlap between cases with *KDELR3* PTVs and ones with diagnostic variants.

4. How is *KDELR3* related to retrograde vesicle mediated transport and pathway that is also found to be enriched with PTVs? The link is missing, and the mechanism needs to be better explained and linked.

Reply - Thank you for the insightful comments. The *KDELR1*, *KDELR2* and *KDELR3* genes are involved in Retrograde vesicle-mediated transport, Golgi to endoplasmic reticulum, which is enriched rare PTV among EM cases. Thus, in our revised manuscript, we have performed gene set burden tests by removing *KDELR1*, *KDELR2*, and *KDELR3* from the members of retrograde vesicle-mediated transport, respectively. And we found only when removed *KDELR3* removed from the pathway, the burden test showed no significant enrichment (OR = 2.23, $P = 0.112$; Supplementary Fig. 8). In addition, we have added some potential mechanism explanations between *KDELR3* and retrograde vesicle-mediated transport in Discussion.

Supplementary Fig. 10. Burden analysis of genes involved in retrograde vesicle-mediated transport, Golgi to ER, when removed *KDELR1*, *KDELR2* or *KDELR3*. PTVs, D-mis, B-mis and synonymous variants within rare variants are displayed.

Minor points

1. Line 232: This is a comparison between three EM cases with *KDELR3* PTV versus three controls. However, the evidence do not support the conclusion. It is obviously there are clinical differences by comparing EM cases to controls. It is difficult to understand how to conclude that *KDELR3* plays a role in such difference based on this comparison. If you randomly chose three EM cases regardless of *KDELR3* status, would you expect a similar or different clinical difference?

Reply - Thank you for your constructive comments. In response to the comments, we have conducted the ocular examination on six EM eyes randomly selected from our cohort. We have found significant differences ($P < 0.05$) in terms of spherical equivalent refraction, axial length, intraocular pressure and subfoveal choroidal thickness between randomly selected EM cases and healthy controls. Although there were no significant differences of ophthalmologic factors between EM cases with *KDELR3* mutation and randomly selected EM cases with the current sample size, we also found obvious differences of subfoveal choroidal thickness between EM cases with *KDELR3* mutation (283.2 μm) and randomly selected EM cases (338.8 μm) (Figure 4b).

2. Line 243: when increasing MO dosage, the phenotype seems to be different or reversed. Any proposed mechanism?

Reply - Thank you for your valuable comments. In recent years, several studies have sought to elucidate the roles of KDELRs in cell physiology, and their possible involvement in development. For instance, in *KDELR* mutant mice, cells are sensitive to ER stress, which finally develop into dilated cardiomyopathy (Hamada, H., et al., 2004). Bi-allelic variants of the *KDELR2* gene have recently been described in patients with osteogenesis imperfecta (OI), a rare heterogeneous connective tissue disorder characterized by susceptibility to bone fractures along with neurodevelopmental disorders (Efthymiou, S., et al., 2021). In the *Drosophila melanogaster*, *KdelR* mutations are early larval lethal, with homozygous mutant animals dying as first instar larvae (Elliott A., et al., 2013). In our study, we also found that with further increases in MO dose, zebrafish develop malformations in body and eye development. We hypothesized that as the *kdelr3*-MO dose increased, the systemic effect was beyond

the tolerance range of zebrafish, resulting in a reversal of the ocular phenotype. Based on these results, we reduced *kdelr3*-MOs concentration and used 0.5ng as the knockdown dose.

3. Line 344: it is premature to call KDELR3 PTV a “pathogenic variant”. The gene-disease relationship and mechanism are not definitive at this time, because the current evidence are mostly from case-control signals and zebrafish model of gene knockdown. The findings are also not replicated in independent cohorts.

Reply - Thank you for your thoughtful advice. We have changed “pathogenic variant” to “potential pathogenic variant” in our revised manuscript.

Reviewer #2

Comments for the Author

*Thank you for the opportunity to review the manuscript authored by Yuan et al., which offers a comprehensive WES genetic analysis concerning extreme myopia (EM). This study not only identifies new genes, particularly *KDELR3*, but also delineates relevant cell types associated with EM.*

*Moreover, it includes a functional evaluation of *KDELR3* within a zebrafish model.*

Reply - We greatly appreciate reviewer #2's recognition to the quality and importance of our work.

While the manuscript adeptly presents these findings, there are issues that warrant attention.

Key aspects for consideration include:

WES analysis:

*The authors included 449 cases and 9,606 controls in their EM WES analysis, with no evidence linking common and low-frequency variants with EM. Given the relatively small sample size, this lack of findings is understandable. Subsequently, a gene-based rare variant analysis was conducted, identifying *KDELR3* and *VNIR4* as significant. In light of the binary classification of EM, it would be beneficial to contextualize these results within the broader spectrum of myopia, refraction error, and high myopia, incorporating both previous findings by the authors and those by other groups.*

Reply - Thank you for your valuable comments. To contextualize these results within the broader spectrum of refraction error, myopia, and high myopia, we further investigated the association of *KDELR3* PTVs with spherical equivalent (SE) in our MAGIC cohorts. (1) We found heterozygous carriers of *KDELR3* PTVs was associated with 0.750 lower SE diopters, and their distribution across SE diopters was drastically shifted toward lower SE diopters in the both eyes ($P = 0.0180$ for right and $P = 0.0004$ for left, respectively, Wilcoxon rank sum test; Fig. 3e). (2) We also performed the Fisher's exact test to find associations between *KDELR3* with rare PTVs and common high myopia ($-10.00 \text{ D} \leq \text{SER} \leq -6.00 \text{ D}$), and found *KDELR3* had nominal significant signals (OR = 2.19, $P = 0.038$). (3) Recently, Karczewski et al determined gene-based association investigating 4529 phenotypes in 394,841 UK Biobank exomes (<https://app.genebase.org/>). We used the published UK Biobank portal to demonstrate phenome-wide association studies (PheWAS) of rare predicted loss-of-function (pLoF) in *KDELR3* on other diseases (Supplementary Fig. 20). There were 4528 pLoF gene burden associations with *KDELR3*, and *KDELR3* was most significant associated with both corneal resistance factor left custom ($P = 4.04 \times 10^{-5}$) and corneal hysteresis left custom ($P = 4.23 \times 10^{-5}$). It is also known that high myopia has lower corneal hysteresis than emmetropes and low myopia.

Supplementary Fig. 16. PheWAS Manhattan Plot. Each data point represents phenotypic associations with *KDELR3* in pLoF model. The data points are grouped and color-coded by phenotype groups (x-axis) and $-\log_{10}(P\text{-value})$ (y-axis).

Cell type enrichment:

The authors employed two methods to associate scRNA-seq with the polygenic burden of rare variants: a cell type level test, which showed no significant enrichment, and a single-cell resolution polygenic burden score (MAGIC, fibroblasts). Considering existing frameworks for cell type enrichment tests (e.g., magma-based, LDSC-based, scDRS), the uniqueness and advantages of the approach used in this study are not clear.

Reply - Many thanks for the insightful comments. To characterize considerable heterogeneity within each cell type, we have applied single-cell enrichment approach MAGMA-based scDRS, to discern EM-associated cell types and subpopulations with ExWAS. We have found none of the genes were significantly associated with the HM in MAGMA analysis (Supplementary Fig. 5) and scDRS did not identify the EM-relevant cells with significant individual cell-level disease-association (Supplementary Fig. 6). These existing methods mainly focused on common variants (e.g., LDSC-SEG, MAGMA-based scDRS and RolyPoly) that cannot differentiate between common and rare variants, due to the limited statistical power for rare variants. Therefore, we introduce the single-cell polygenic burden score (scPBS) to evaluate polygenic burden enrichment of rare variants in individual cells of scRNA-seq data. For rare variants alone, scPBS successfully identified $PI16^+/SFRP4^+$ fibroblasts as the top-rank tissue for the EM. scPBS offers a statistical framework to identify single-cell specific enrichment signals attributed to rare variants.

Supplementary Fig. 5. Manhattan plot of MAGMA gene-based genetic association analysis on EM. Bonferroni significance threshold is $0.05/20,000 = 2.5 \times 10^{-6}$

Supplementary Fig. 6. Associations of single-cell of human eyes with EM. (a) Uniform manifold approximation and projection (UMAP) embedding plot shows the cellular component of a scRNA-seq dataset for human embryonic eyes. Abbreviations are as follows: RPCs, retinal progenitor cells; NPCa, neurogenic RPCs; MGCs, Müller glial cells; HCs, horizontal cells; BCs, bipolar cells; ACs, amacrine cells; RGCs, retinal ganglion cells; RPE, retinal pigment epithelia; Endothelium, endothelial cells. (b) Subpopulations of cell-type associated with EM. The color denotes scDRS disease scores. (c) For each cell based on the empirical distribution of the pooled normalized control scores across all

control gene sets and all cells. (d) Heatmap colors for each cell type-disease pair denote the proportion of significant associated-cells.

Reviewer #3

Comments for the Author

The manuscript by Yuan et. al. describes the identification of endoplasmic reticulum protein retention receptor 3, KDELR3, as being a novel gene involved in the progression of extreme myopia (EM) whereby spherical equivalent refractive error (SER) is characterised as < -10.00 D. Determining the causative factors involved in the progression to EM is imperative in curbing the increasing health and socio-economic burden resulting from vision loss. The authors have shown compelling and thorough evidence that identifies KDELR3 rare protein truncating variants (PTVs) in patients suffering EM by using a range of techniques such as whole-exome sequencing (WES), single-cell RNA-sequencing (scRNA-seq), and bulk-cell RNA-sequencing (bcRNA-seq). This study postulates that KDELR3 expressed in scleral fibroblast are responsible for scleral extracellular matrix (ECM) organization and the regulation of both scleral thickness and strength. The authors have also investigated the function of KDELR3 in zebrafish larvae by generating the first transient knockdown (KD) model of KDELR3 using splice-modifying morpholino (MO) anti-sense oligonucleotides. The transient KD of KDELR3 resulted in an eye phenotype consistent with myopia, increased axial length and lens diameter. The authors have conducted extensive genetic and molecular studies to investigate the modulation of expression in genes downstream of KDELR3.

This original work provides a framework for further investigation into the molecular signalling of KDELR3 and the role it plays within ECM organization and the modulation of scleral thickness in EM. As highlighted by the authors, there is considerable evidence to suggest genetics primarily drive the succession of extreme myopia and as such it is unlikely increased environmental risk factors such as near-work are major contributing factors. As a result, it remains imperative to conduct large-scale population genetic screening to further investigate the genetic origin of EM. The work conducted by Yuan et. al. is coherent and supports the conclusions of the study and has contributed to the catalogue of known EM-associated genes.

Reply - We greatly appreciate the time and effort reviewer #3 have invested in evaluating our work and for the thoughtful suggestions.

The methodology and statistical analysis are sound and meets the expected standard in the field and is detailed enough to be reproduced. Some questions relating to the zebrafish MO model are listed below:

-As the authors are the first to describe a zebrafish KDELR3 splice modifying MO model, can they confirm KDELR3 KD up to 5 dpf using PCR and sequencing to identify any inclusion of introns or

frameshifts? The target exon is essential in generating splice modifying MOs. Is it possible to include this in supplementary figures as it is important to validate new MOs.

Reply - Thank you for your essential comments. In revised manuscript, we have provided the RT-PCR results from 3dpf to 5dpf with electrophoresis analysis of RT-PCR products (Supplementary Fig. 11), which indicate that the knockdown of *kdelr3*-MO is effective and is enabled to maintain up to 5 dpf. In addition, we have used both SUPPA2 and custom analysis scripts to identify transcripts associated with MO-induced skipped exon in our RNA-seq datasets from *kdelr3*-MO zebrafish eyeballs, and have found that *kdelr3*-MO zebrafish exhibited aberrant use of exons compared with the control zebrafish eyeballs (Supplementary Fig. 12 and Supplementary Table16).

Supplementary Fig. 13. RT-PCR confirmed expression change in *kdelr3*-MO injected zebrafish larvae. 0.5 ng of *kdelr3*-MOs were tested at 3dpf, 4dpf and 5dpf. 0.5 ng of control-MOs were tested at 5dpf.

being the most effective and exhibiting the most biologically specific phenotype. The doses used by the authors are much lower than this range, using 0.15 – 0.5 ng, can the authors comment on why such low doses were used? Calibration of delivery in each experiment is critical.

Reply - Thank you for these constructive comments. In recent years, several studies have sought to elucidate the roles of KDELRs in cell physiology, and their possible involvement in development. For instance, in *KDELR* mutant mice, cells are sensitive to ER stress, which finally develop into dilated cardiomyopathy (Hamada, H., et al., 2004). Bi-allelic variants of the *KDELR2* gene have recently been described in patients with osteogenesis imperfecta (OI), a rare heterogeneous connective tissue disorder characterized by susceptibility to bone fractures along with neurodevelopmental disorders (Efthymiou, S., et al., 2021). In the *Drosophila melanogaster*, *KdelR* mutations are early larval lethal, with homozygous mutant animals dying as first instar larvae (Elliott A., et al., 2013). We also found MO dosage was further increased, the zebrafish showed develop malformations in body development. Based on these results, we reduced *kdelr3*-MOs concentration and used 0.5ng as the knockdown dose. In recent years, several studies also chose 0.5ng as the appropriate knockdown dose in zebrafish (S. Javad Rasouli & Didier Y. R. Stainier, 2017; Wing W., et al., 2017; Jason L., et al., 2019).

To further verify that the ocular phenotype found in *kdelr3*-deficient morphants is not due to off-target effects, we performed mRNA compensation rescue experiments in rescue experiments. We found that the phenotype of the *kdelr3* knockdown is reversible after using full-length mRNA compensation in *kdelr3*-deficient morphants (Supplementary Fig. 15).

Supplementary Fig. 15. Quantification of eye axial length, the axis-to-body length ratio, lens diameter and lens-to-body length ratio for *kdelr3*-deficient and rescue zebrafish. Data was analyzed by Student's t test, * $P < 0.05$, ** $P < 0.01$, *** $P < 0.001$ significantly different from control 0.5 ng group.

-ZFIN currently has not described expression of *KDEL3* in the visual systems. Perhaps you could update this atlas.

Reply - Thank you for your thoughtful advice. After publication of the manuscript, we will submit the expression of *KDEL3* in the visual system to the ZFIN database.

-Can the authors use SS-OCT imaging to calculate axial length and lens diameter in the MO model, normalised to body length?

Reply - We sincerely apologize that we can't use SS-OCT imaging to calculate axial length and lens

diameter in the MO model. Since the SS-OCT instrument for zebrafish is not yet widely available, it is not yet available at our institute. Zebrafish were anesthetized with tricaine and imaged from lateral and dorsal perspectives using a stereoscopic fluorescence microscope (SZX16, OLYMPUS, Japan). The measurement of axial length (from the front of the cornea to the back of the sclera) and lens diameter (from the anterior to the posterior surface of the lens) from histological cross sections used in this study was reported in several studies (Veth KN. et al., 2011; Liu S. et al., 2022; Huang X. et al., 2018), which could be a suitable alternative to SS-OCT imaging.

Minor comments:

-Figure titles missing under the primary figures.

Reply - Thank you for the careful checking. We have added figure titles under the primary figures.

-Do the authors plan to generate a KO line to further investigate the role of KDELR3?

Reply - Thank you for your insightful advice. To investigate the mechanism of KDELR3 induced EM, we plan to construct KO cell lines based on RPE cells and conditional knockout mouse models targeting the *KDELR3* gene in the eyes. These models will present progressive changes in longitudinal measurements and refractive error.

-A clearer Western blot showing decreased COL1a1 protein levels in the KDELR3 shRNA sample would be more convincing as the one shown suggests a problem with transfer.

Reply - Many thanks for this helpful suggestion. We have provided a clearer Western blot in figure 5 and Supplementary Fig. 19 to show decreased *COL1a1* protein levels in the KDELR3 shRNA sample in our revised manuscripts.

Supplementary Fig. 19. Western blot analysis. a, Protein levels of scleral KDEL3, COL1 α 1, and α -SMA were detected by Western blot in KDEL3-deficient HSF cells. b, Protein levels of scleral KDEL3, COL1 α 1, and α -SMA in RPE cells.

-Supplementary Figure 13 is named Supplementary Fig. 1

Reply - Thank you for your thoughtful advice. We have revised the ordering of figures in our revised manuscripts

-Supplementary Figure 14 is named Supplementary Fig. 2

Reply - Thank you for your thoughtful advice. We have revised the ordering of figures in our revised manuscripts.

REVIEWERS' COMMENTS

Reviewer #1 (Remarks to the Author):

The authors have addressed my concerns.

Reviewer #2 (Remarks to the Author):

I have no additional comments.

Reviewer #3 (Remarks to the Author):

Noteworthy results and significance to the field:

The manuscript by Yuan et. al. describes the identification of endoplasmic reticulum protein retention receptor 3, KDELR3, as being a novel gene involved in the progression of extreme myopia (EM) whereby spherical equivalent refractive error (SER) is characterised as < -10.00 D. Determining the causative factors involved in the progression to EM is imperative in curbing the increasing health and socio-economic burden resulting from vision loss. The authors have shown compelling and thorough evidence that identifies KDELR3 rare protein truncating variants (PTVs) in patients suffering EM by using a range of techniques such as whole-exome sequencing (WES), single-cell RNA-sequencing (scRNA-seq), and bulk-cell RNA-sequencing (bcRNA-seq). This study postulates that KDELR3 expressed in scleral fibroblast are responsible for scleral extracellular matrix (ECM) organization and the regulation of both scleral thickness and strength. The authors have also investigated the function of KDELR3 in zebrafish larvae by generating the first transient knockdown (KD) model of KDELR3 using slice-modifying morpholino (MO) anti-sense oligonucleotides. The transient KD of KDELR3 resulted in an eye phenotype consistent with myopia, increased axial length and lens diameter. The authors have conducted extensive genetic and molecular studies to investigate the modulation of expression in genes downstream of KDELR3.

Revision report:

I am satisfied that the authors have addressed my previous comments regarding methodology and results, and have kindly included more datasets in their manuscript as a result. I think this is an important contribution to the field and I wish the authors all the best for their research.

Point-to-point response to the reviewers (NCOMMS-23-50564A)

Reviewer #1

Comments for the Author

The authors have addressed my concerns.

Reply - We greatly appreciate reviewer #1's recognition of the quality and importance of our work.

Reviewer #2

Comments for the Author

I have no additional comments.

Reply - We greatly appreciate reviewer #2's recognition to the quality and importance of our work.

Reviewer #3

Comments for the Author

Noteworthy results and significance to the field: The manuscript by Yuan et. al. describes the identification of endoplasmic reticulum protein retention receptor 3, KDELR3, as being a novel gene involved in the progression of extreme myopia (EM) whereby spherical equivalent refractive error (SER) is characterised as < -10.00 D. Determining the causative factors involved in the progression to EM is imperative in curbing the increasing health and socio-economic burden resulting from vision loss. The authors have shown compelling and thorough evidence that identifies KDELR3 rare protein truncating variants (PTVs) in patients suffering EM by using a range of techniques such as whole-exome sequencing (WES), single-cell RNA-sequencing (scRNA-seq), and bulk-cell RNA-sequencing (bcRNA-seq). This study postulates that KDELR3 expressed in scleral fibroblast are responsible for scleral extracellular matrix (ECM) organization and the regulation of both scleral thickness and strength. The authors have also investigated the function of KDELR3 in zebrafish larvae by generating the first transient knockdown (KD) model of KDELR3 using slice-modifying morpholino (MO) anti-sense oligonucleotides. The transient KD of KDELR3 resulted in an eye phenotype consistent with myopia, increased axial length and lens diameter. The authors have conducted extensive genetic and molecular studies to investigate the modulation of expression in genes downstream of KDELR3.

Revision report: I am satisfied that the authors have addressed my previous comments regarding methodology and results, and have kindly included more datasets in their manuscript as a result. I think this is an important contribution to the field and I wish the authors all the best for their research.

Reply - We greatly appreciate the time and effort reviewer #3 have invested in evaluating our work.